# Master corepressor inactivation through multivalent SLiM-induced polymerization mediated by the oncogene suppressor RAI2

Nishit Goradia[1,8], Stefan Werner [2,3,8], Edukondalu Mullapudi[1], Sarah Greimeier[2], Lina Bergmann [2], Andras Lang[4], Haydyn Mertens[1], Aleksandra Węglarz[2], Simon Sander [2], Grzegorz Chojnowski[1], Harriet Wikman [2], Oliver Ohlenschläger [4], Gunhild von Amsberg[5,6], Klaus Pantel [2] ✉ & Matthias Wilmanns [1,7] ✉

While the elucidation of regulatory mechanisms of folded proteins is facilitated due to their amenability to high-resolution structural characterization, investigation of these mechanisms in disordered proteins is more challenging due to their structural heterogeneity, which can be captured by a variety of biophysical approaches. Here, we used the transcriptional master corepressor CtBP, which binds the putative metastasis suppressor RAI2 through repetitive SLiMs, as a model system. Using cryo-electron microscopy embedded in an integrative structural biology approach, we show that RAI2 unexpectedly induces CtBP polymerization through filaments of stacked tetrameric CtBP layers. These filaments lead to RAI2-mediated CtBP nuclear foci and relieve its corepressor function in RAI2-expressing cancer cells. The impact of RAI2-mediated CtBP loss-of-function is illustrated by the analysis of a diverse cohort of prostate cancer patients, which reveals a substantial decrease in RAI2 in advanced treatment-resistant cancer subtypes. As RAI2-like SLiM motifs are found in a wide range of organisms, including pathogenic viruses, our findings serve as a paradigm for diverse functional effects through multivalent interaction-mediated polymerization by disordered proteins in healthy and diseased conditions.

Imbalanced transcription of a large number of genes can be caused at various levels, ranging from alterations in individual transcription factors to dysfunctional corepressor and coactivator complexes[1]. C-terminal binding proteins (CtBPs) are ubiquitous master transcriptional coregulators associated with the Polycomb Repressive Complex 2 (PRC2)[2,3], with essential functions in the development and oncogenesis of various tumor entities, including breast cancer (BC) and prostate cancer (PC)[4,5]. CtBPs can affect cancer progression and promote cell survival, proliferation, motility, and the epithelial-mesenchymal transition (EMT) by repressing many tumor suppressor genes such as *CDKN1A*, *CDH1*, or *BRCA1*[6–9].

[1]European Molecular Biology Laboratory, Hamburg Unit, Notkestrasse 85, 22607 Hamburg, Germany. [2]University Medical Center Hamburg-Eppendorf, Department of Tumor Biology, University Cancer Center Hamburg, Martinistrasse 52, 20246 Hamburg, Germany. [3]University Medical Center Hamburg-Eppendorf, Mildred Scheel Cancer Career Center HaTriCS4, Martinistrasse 52, 20246 Hamburg, Germany. [4]Leibniz Institute on Aging, Fritz-Lipmann-Institute, Beutenbergstraße 11, 07745 Jena, Germany. [5]Martini Clinic, Martinistrasse 52, 20246 Hamburg, Germany. [6]Department of Hematology and Oncology, University Medical Center Hamburg-Eppendorf, Hamburg, Germany. [7]University Medical Center Hamburg-Eppendorf, Martinistrasse 52, 20246 Hamburg, Germany. [8]These authors contributed equally: Nishit Goradia, Stefan Werner. ✉e-mail: pantel@uke.de; matthias.wilmanns@embl-hamburg.de

In humans, two closely related proteins CtBP1 and CtBP2 are encoded by two paralogous genes *CTBP1* and *CTBP2*, respectively[10]. Both CtBPs share a conserved three-domain arrangement in common that includes a split binding domain interacting with PxDLS sequence motif-containing proteins, a central dehydrogenase domain, and a C-terminal domain, which is less conserved in the two CtBP paralogues. The binding of NAD⁺/NADH to the respective dehydrogenase domain is required for CtBP dimerization, which allows further NAD⁺/NADH -independent assembly into active CtBP tetramers as part of large corepressor complexes for recruitment to multiple chromatin-binding sites[6,11–13].

Although a single PxDLS-like short linear motif (SLiM) on a target protein sequence is sufficient for CtBP binding[14–16], it was intriguing to detect the presence of two such repeated non-canonical motifs on the Retinoic Acid-Induced 2 (RAI2) protein[17]. RAI2 has been characterized as a potential metastasis suppressor protein that inhibits the dissemination of tumor cells to the bone marrow and when absent, induces an aggressive tumor phenotype in BC cells[17,18]. Hence, the potential antagonistic roles of RAI2 and CtBP as tumor suppressor and oncogene, respectively, could be explained through their ability to bind directly to each other.

Here, we demonstrate that the presence of a twofold repeated RAI2 ALDLS motif sequence segment leads to well-defined CtBP polymerization at the molecular level and nuclear foci at the cellular level. We further show that the presence of RAI2 leads to an epigenetically regulated relief of CtBP-mediated repression of the key tumor suppressor gene *CDKN1A* through a reduction in histone 3 trimethylation (H3K27me3). Consistent with these findings, we reveal that the emergence of neuroendocrine traits in a relevant prostate tumor cell line model and PC progression to treatment-resistant stages in metastatic PC patient samples are both associated with the loss of RAI2. Ultimately, our data may serve as a paradigm for the inactivation of other protein factors by polymerization and could provide molecular insights into the development of therapeutic resistance.

## Results

### CtBP binding to the tandem motif-containing RAI2 is autonomous

As RAI2 remarkably contains two CtBP-binding motifs of identical sequence (ALDLS) at RAI2 sequence positions 316–320 and 342–346, respectively, thus separated by a short 21-residue linker only, we first investigated whether there is preferential binding by one of the two ALDLS motifs through a specific protein environment and whether there is any interference between the two of them. As the RAI2-binding domain is conserved between CtBP1 and CtBP2 sequences, we treated them analogously in this study. First, we analyzed the molecular properties of the two binding partners RAI2 and CtBP1/2 individually. While the structural basis of the tetrameric CtBP arrangement has already been established[19], there are no experimental structural data for RAI2 due to the lack of any defined fold predictions (Supplementary Fig. 1). We established purification protocols for three truncated versions of RAI2 constructs with both ALDLS motifs intact, designated RAI2(WT, 303–530), RAI2(WT, 303–465) and RAI2(WT, 303–362) (Fig. 1a, Supplementary Fig. 2a). Circular dichroism (CD) spectroscopy and small angle X-ray scattering (SAXS) data confirmed that RAI2 is mostly intrinsically disordered (Fig. 1b, c, Supplementary Figs. 2b, 3 and 4, Supplementary Data 1 and 2). In addition, nuclear magnetic resonance (NMR) spectroscopy chemical shift analysis of RAI2 (303-362) revealed typical random coil values throughout the entire sequence of this protein fragment (Fig. 1e, f, Supplementary Fig. 5a–c, Supplementary Data 3). Evaluation of the molecular flexibility (Supplementary Fig. 5d) revealed that all RAI2 residues exhibited a HN bond mobility on the fast time scale, which is reflected by negative $^{15}N$-{$^{1}H$}-NOE values.

Both RAI2(303–362) variants with one of the two ALDLS sites impaired (M1, M2) bind with a dissociation constant of approximately

5 μM (Fig. 1d, Supplementary Table 1). They showed no gain in binding affinity when compared to previous data on synthetic peptides[17]. RAI2(WT) with both CtBP-binding sites intact displayed only a modest fivefold increase in binding affinity, rendering any cooperative effect of the two binding sites acting in tandem to be unlikely (Fig. 1d, Supplementary Table 1). We also detected an approximately twofold change in binding stoichiometry, demonstrating that both RAI2 ALDLS sites are accessible to CtBP1 in parallel. The binding affinity did not increase further when longer RAI2 constructs (WT, 303–465, 303–530) were used, demonstrating that the nature of the RAI2/CtBP1 interaction through twofold repeated SLiMs from RAI2 does not depend on other parts of RAI2 beyond the sequence of the smallest RAI2 construct (WT, 303–362) used for these experiments (Supplementary Table 1). Based on these findings, we used either RAI2(WT, 303–362) or RAI2(WT, 303–465) in all subsequent biophysical and structural experiments. CD and SAXS data of the respective CtBP1/RAI2 complexes revealed a partially folded protein content, which is expected for mixed complexes of folded and unfolded protein partners (Fig. 1b, c, Supplementary Figs. 3, 4, and 6a).

To gain insight into the molecular basis of this interaction, we titrated purified CtBP1[20] into RAI2(WT, 303-362) as well as the two RAI2 M1 and M2 variants and characterized the interactions on a single residue basis by NMR spectroscopy (Fig. 1f–h). The marked decrease in the chemical shift intensities of these RAI2 variants upon CtBP1 complex formation demonstrated similar CtBP-binding properties of the two RAI2 ALDLS motifs (Fig. 1h middle and bottom panels, Supplementary Data 4). In contrast to the single ALDLS-containing RAI2 variants, we found a bilobal intensity decrease distribution around the ALDLS motifs in RAI2(WT, 303-362), confirming that CtBP can bind to both RAI2 ALDLS sites simultaneously (Fig. 1h top panel, Supplementary Data 4).

### Tandem ALDLS motif-containing RAI2 induces CtBP oligomerization

Given that both protein binding partners have multivalent binding sites for each other, four in CtBP and two in RAI2, we wondered about the possibility of assembly-mediated oligomerization. As evident from analytical size exclusion chromatography (SEC) profiles, titration of RAI2(WT, 303–465) to CtBP1 resulted in the formation of a high molecular weight complex, indicative of higher order oligomerization (Fig. 2a, Supplementary Fig. 6b, Supplementary Table 2). In contrast, the addition of the M1 and M2 variants to CtBP1 resulted in an increase in retention volume relative to the RAI2(WT, 303–465)/CtBP elution profile, suggesting non-oligomerized, lower molecular weight CtBP1/RAI2 complexes (Fig. 2a, Supplementary Table 2). As expected, there was no change in the CtBP1 elution profile when titrated with the RAI2 M1+M2 variant (Supplementary Fig. 6c).

To further investigate the nature of this high molecular weight CtBP1/RAI2 complex with both ALDLS binding sites intact, we used SAXS to assess its maximum particle dimension ($D_{max}$) in solution. When RAI2(303-465) M1 or M2 variants were added to CtBP1, we observed a moderate increase of the CtBP1 particle size from 14 nm to approximately 18 nm (Fig. 2b, Supplementary Data 5). However, when RAI2(WT, 303-465) was added to CtBP1, the average particle size doubled from 14 nm to 28 nm, consistent with an elongated assembly of both protein components. These observations also held true when shorter RAI2(303-362) constructs were titrated onto CtBP1 (Supplementary Figs. 3, 4 and 6d–f, Supplementary Data 6).

To understand the implications of this interaction in a cellular setting, we analyzed CtBP1/RAI2 assembly in KPL-1 BC cells[17] using confocal laser scanning microscopy. In the presence of RAI2(WT), we detected distinct CtBP1/RAI2 nuclear foci. The number and volume of these foci were significantly reduced when RAI2 variants (M1, M2, M1+M2), in which one or both ALDLS CtBP-binding sites were mutated, were used (Fig. 2c, d). Our findings for CtBP2/RAI2 assembly

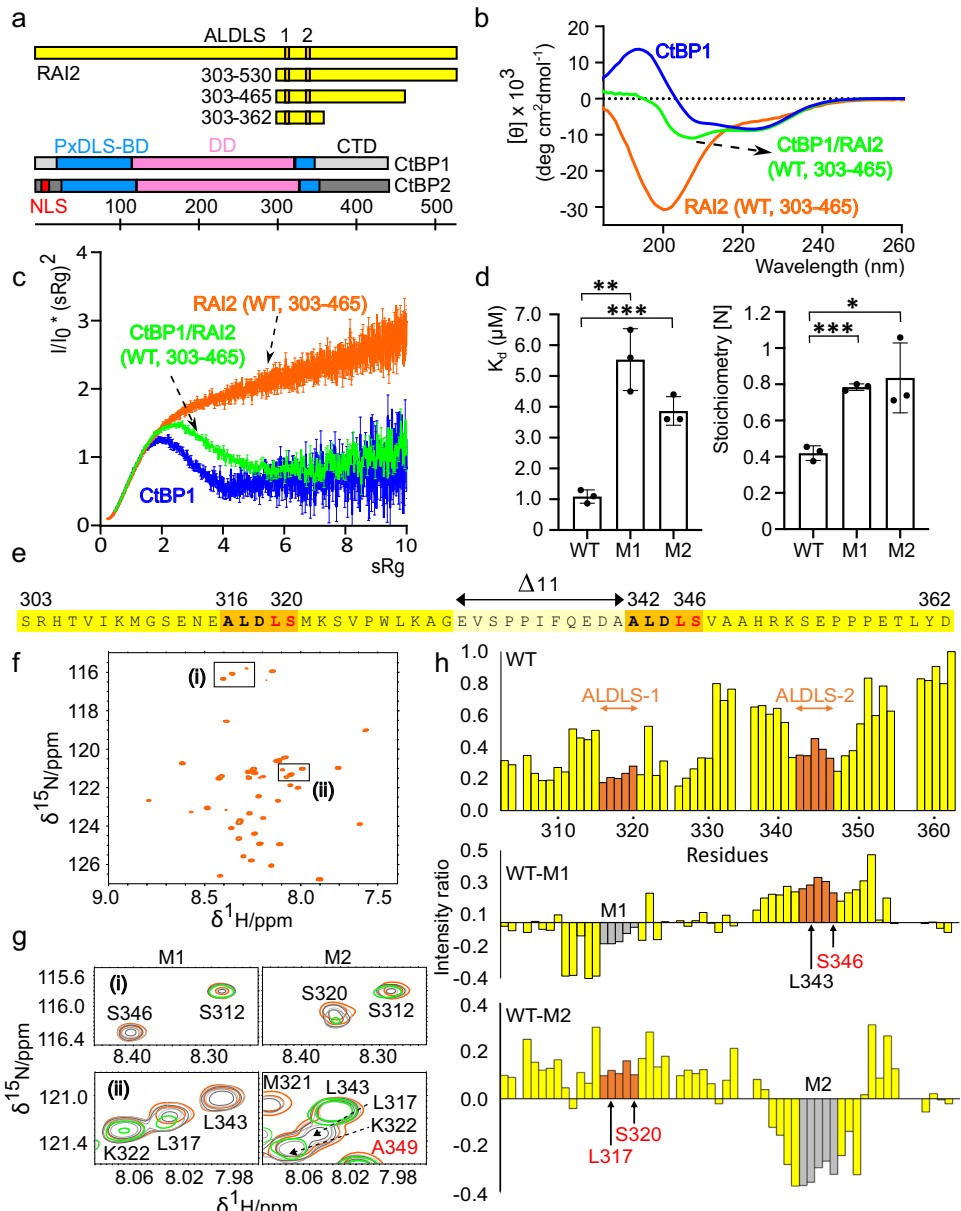

**Fig. 1 | Tandem ALDLS motif-containing RAI2 binding to CtBP is autonomous.**
**a** Protein constructs in proportion to their length. Colors: RAI2, yellow: ALDLS motifs 1 and 2, orange; CtBP1 and CtBP2: split PxDLS binding domain (BD), blue; dehydrogenase domain (DD), pink; C-terminal domain (CTD), gray; nuclear localization sequence (NLS), red. CTD gray shades of CtBP1 and CtBP2 indicate CTD diversity. **b** CD spectra of CtBP1(WT, FL) (blue), RAI2(WT, 303–465) (orange) and CtBP1(WT, FL)/RAI2(WT, 303–465) complex (green). **c** SAXS Kratky plots of the protein constructs used in (**b**) indicating the level of structural flexibility and fold content. The plots are based on a single experiment each and the error bars represent the propagated SEM of the scattering intensities measured at each point following Poisson counting statistics[90]. The data have been taken from SASBDB entries SASDQW5, SASDQ46, and SASDQC6. **d** Dissociation constants $K_D$ (left panel) and stoichiometry values (right panel) of RAI2(303–362) WT, M1 and M2 to CtBP1. The data are represented as mean of n = 3 independent biological replicates ± SD. Unpaired t-test was used to evaluate statistical significance applying p

values for a two-sided confidence interval as follows: p < 0.001 (***), p < 0.01 (**), p < 0.05 (*). Definition p values: $K_D$: p = 0.0017 M1, p = 0.0007; Stoichiometry: p = 0.0001 M1, p = 0.0217 M2. **e** Sequence of RAI2(WT, 303–362, yellow) comprising the twofold repeated ALDLS motif (orange). Residues mutated in RAI2 M1, M2 and M1 + M2 variants are in red. Removed linker residues in RAI2(Δ11, 303–362) are in light yellow and are indicated by a double arrow. **f** HSQC spectra of RAI2(WT, 303–362). **g** Effects of CtBP1 titration on selected RAI2 residues (boxed and labeled in **F**) in M1 (left) and M2 (right) variants. Colors of increasing amounts of titrated CtBP1 (WT, 28–353) on RAI2(303–362) variants: orange (1:0), dark gray (1:0.4), light gray (1:0.7), and green (1:1). **h** Histogram of chemical shift perturbation intensity ratios upon titration of CtBP1(28–353) onto RAI2(303–362) variants (WT, M1, M2). Mutated ALDLS sites are in gray. Low intensity ratios indicate perturbation of RAI2 residues in the presence of CtBP. See also Supplementary Figs. 1–5, Supplementary Table 1 and Supplementary Data 1, 3–5, and 8. Source data are provided as a Source Data file.

are consistent with those observed for CtBP1/RAI2 (Supplementary Fig. 7a–c).

## Molecular Basis of RAI2-induced CtBP oligomerization
Next, we used high-resolution structural biology approaches to elucidate the molecular basis of RAI2-mediated CtBP oligomerization.

To define a reference module for RAI2-mediated oligomerization, we first determined the high-resolution crystal structure of CtBP2 in the presence of a single CtBP-binding site RAI2(M2, 303-465) variant (Fig. 2e, Supplementary Fig. 8, Supplementary Table 3). The structure comprises a CtBP/RAI2 complex with 4:4 stoichiometry forming a flat X-shaped arrangement with overall dimensions of 9.5 × 9.5 × 5 nm

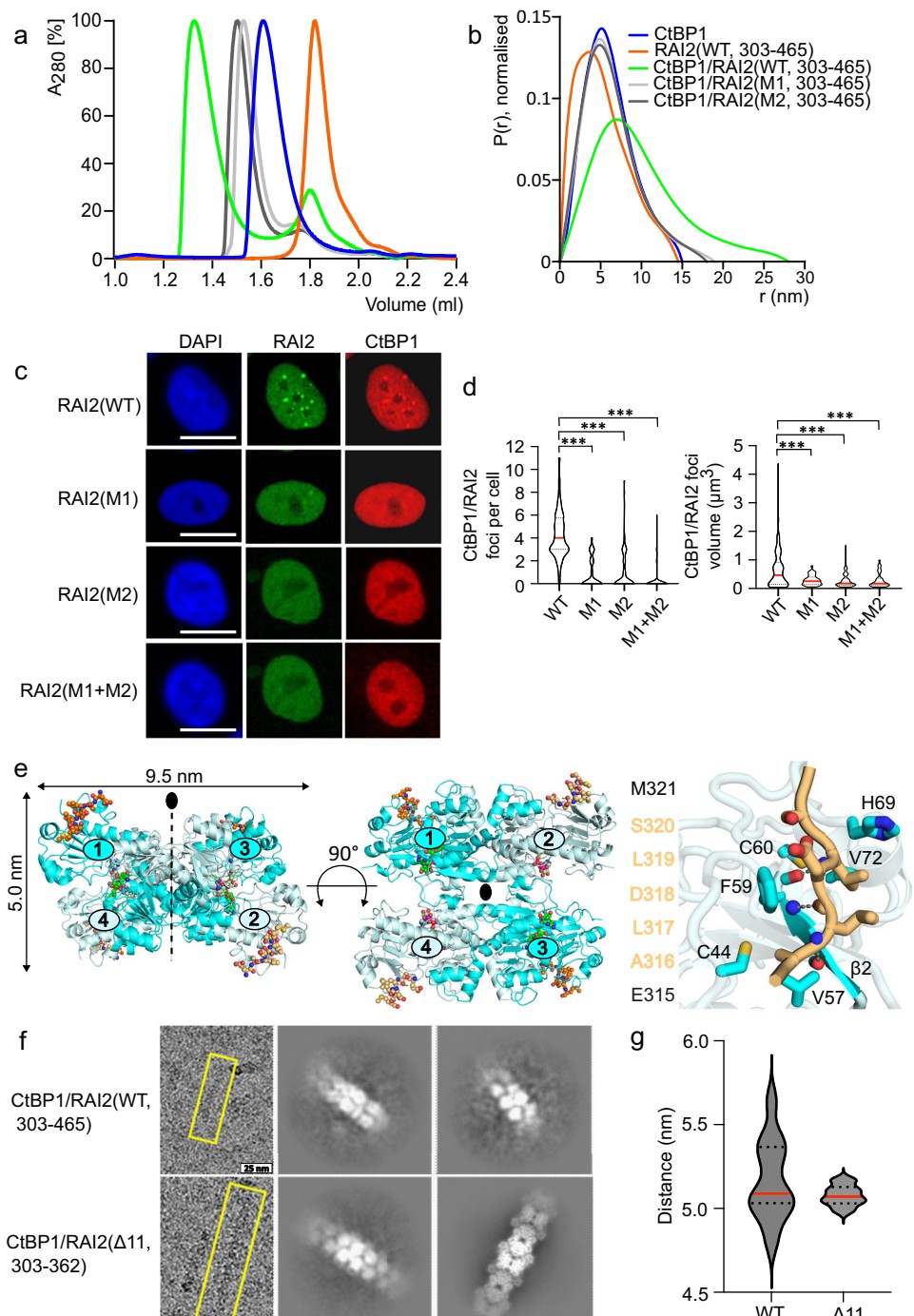

(Fig. 2e, left and middle panel). Apart from RAI2 residues 315–321, which directly interact with CtBP2 (Fig. 2e, right panel, Supplementary Fig. 8), there was no interpretable electron density for the remaining RAI2 sequence, confirming that it remains unfolded upon assembly with CtBP.

We next investigated a molecular mechanism for RAI2-induced CtBP oligomerization using cryogenic-electron microscopy (cryo-EM), which revealed that the interaction of CtBP1 with RAI2 results in extended polymeric structures with repeating structural units at 5 nm intervals (Fig. 2f, g). These spacings correspond to the thickness of individual CtBP1 tetramers (Fig. 2e). However, any further high-resolution structural analysis was not possible due to the inherent flexibility and structural disorder of these fibers.

To gain structural insight into RAI2-mediated CtBP polymerization at high resolution, we shortened the RAI2 linker connecting the two ALDLS CtBP-binding sites by 11 residues (331–341) (Fig. 1e). The resulting RAI2 variant (Δ11, 303–362) retained the ability to bind CtBP1 with comparable affinity, stoichiometry, and particle size, as well as the ability to form nuclear foci (Supplementary Tables 1 and 2, Supplementary Fig. 9a–d). As evident from cryo-EM 2D classes, we found a well-defined filament that lacked the high flexibility of the CtBP1(WT)/RAI2(WT, 303–362) complex (Fig. 2f, g), allowing the determination of its structure at 3.0 Å resolution (Fig. 3, Supplementary Figs. 10–12, Supplementary Table 4). The structure revealed a RAI2-mediated CtBP1 filament with stacked tetrameric CtBP1 assemblies, explaining the layers observed at 5 nm spacing for both RAI2(WT) and RAI2(Δ11)-mediated CtBP1 polymers (Fig. 2f, g). For structural refinement, we

**Fig. 2 | Tandem ALDLS motif-containing RAI2 induces CtBP polymerization.**
**a** Normalized SEC profiles of CtBP1(WT, blue), RAI2(WT, 303–465, orange) and
complexes of CtBP1(WT) with RAI2(WT, green), RAI2(M1, light gray), RAI2(M2, dark
gray). **b** SAXS distance distribution P(r) profiles of CtBP1 in complex with
RAI2(303–465) variants. The plots are based on a single experiment and the data
have been taken from SASBDB entries SASDQW5, SASDQ46, SASDQC6, SASDQD6,
and SASDQE6. **c** Immunofluorescence staining of CtBP1 (red) and RAI2 variants
(green) in KPL-1 cell nuclei. Nuclei are stained with DAPI (blue). Scale bars: 10 μm.
**d** Violin plot depicting the number of CtBP1/RAI2 foci per cell (left) and foci
volumes (right) of one representative experiment. Medians and quartiles have been
shown as red and dashed black lines, respectively. The foci per cell were plotted
based on 100 cells for each RAI2 construct used. Foci volumes were plotted based
on n = 153 (WT), n = 38 (M1), n = 85 (M2), n = 39 (M1 + M2) foci analyzed. Unpaired
t-test was used to evaluate statistical significance by applying p values for a two-
sided confidence interval as follows: $p < 0.001$ (***), $p < 0.01$ (**), $p < 0.05$ (*). Defi-
nition p values: foci per cell and foci volumes = p < 0.001. RAI2(WT) was used as a
reference. **e** Crystal structure of the tetrameric [(CtBP2(WT, 31–364)/NAD⁺)₂]₂/
RAI2(M2, 303–465)₄ complex in side view (left) and top view (middle). CtBP2

molecules are in cartoon presentation, colored cyan, and pale cyan, and numbered.
The visible segments of RAI2 (315–321, orange) and NAD⁺/NADH ligands (green) are
in ball-and-stick representation. Oxygen, nitrogen, and phosphorus atoms are in
red, blue, and magenta, respectively. Right panel, zoom of the CtBP/RAI2-binding
site, highlighting specific CtBP-RAI2 interactions within 4 Å distance. The visible
RAI2 sequence around the interacting ALDLS motif is shown on the left. Hydrogen
bonds are indicated by dashed lines. The CtBP strand β2 is highlighted and labeled
(*cf.* Supplementary Fig. 13). **f** Cryo-EM micrographs (left panel) and 2D classes of
CtBP1(WT, FL)/RAI2(WT, 303–465) and CtBP1(WT, FL)/RAI2(Δ11, 303–362) com-
plexes. Yellow boxes (left) show a representative fiber for each complex. **g** Violin
plot depicting CtBP1 tetramer layer distances (nm), showing reduced distance
variability for RAI2(Δ11) compared to RAI2(WT) used for filament formation with
CtBP1. The distances were obtained from n = 12 cryo 2D classes for each RAI2
construct used. Medians and quartiles are shown in red and black dashed lines,
respectively. See also Supplementary Figs. 3, 4, and 6–9, and Supplementary
Tables 2–4 and Supplementary Data 1 and 5. Source data are provided as a Source
Data file.

restricted our analysis to a fiber consisting of six CtBP1 layers, with
overall dimensions of 6 × 5 nm = 30 nm in length and 9.5 nm in thick-
ness (Fig. 3a), which is consistent with the overall dimensions of tet-
rameric CtBP2 determined by X-ray crystallography (Fig. 2e). In this
structure, the CtBP1/RAI2 fiber is assembled by a 3-fold axis that
defines the longitudinal fiber axis and coincides with one of the 2-fold
axes of the CtBP1 tetrameric assembly (Figs. 2e and 3a). This arrange-
ment is rotated by 120 degrees and translated by 5 nm for adjacent
CtBP1 tetrameric layers, adding a 3₁ screw component to the 3-fold
filament axis. Thus, the minimal unique unit of the RAI2-mediated
filament consists of three tetrameric CtBP1 layers. Despite the con-
siderable size of the surface interfaces of adjacent CtBP tetrameric
layers in the order of 1600 Å², they are almost devoid of any specific
interactions, except for isolated hydrogen bonds involving residues
K46 and R336 with the next CtBP layer (Fig. 3b, c). This may explain
why CtBP filament formation requires the presence of RAI2 as a poly-
merization mediator.

Within this arrangement, we observed two RAI2 peptides per
CtBP1/CtBP1 layer interface that directly connects the RAI2-binding
sites of adjacent CtBP1 tetrameric layers via their two ALDLS motifs 1
and 2 (Fig. 3b, c, Supplementary Fig. 11c), which is consistent with the
N = 0.5 stoichiometry observed by isothermal titration calorimetry
(ITC) (Fig. 1d, right panel). Both RAI2/CtBP-binding sites are formed by
a parallel β-sheet interaction involving β-strand 2 of each CtBP1 RAI2-
binding domain and the central LDL sequence of each of the two RAI2
ALDLS motifs 1 and 2 (Fig. 3d, Supplementary Figs. 11c and 13). This
finding is consistent with our X-ray crystallography data of tetrameric
CtBP2(31-364)/RAI2(M2, 303-465) variant assembly (Fig. 2e right panel,
Supplementary Fig. 8). We did not detect any other specific interac-
tions of flanking residues of the LDL-motif, which explains the rela-
tively moderate binding affinity of the RAI2 interaction with
CtBP1 (Fig. 1d).

To test whether our structural findings on stacked tetrameric
CtBP1 layers explain RAI2-mediated CtBP polymerization, we mutated
one of the few CtBP1 residues (K46) involved in the limited CtBP1/
CtBP1 layer interactions (Fig. 3c) to tryptophan. The binding affinity of
CtBP1(K46W) with RAI2 was significantly decreased, similar to values
observed for CtBP1(WT) and RAI2(M1) or RAI2(M2) mutants (Supple-
mentary Table 1), which could indicate reduced polymerization and
fragmented CtBP/RAI2 fiber formation. In addition, we observed a
decrease in the RAI2/CtBP1 stoichiometry upon binding to values
below 0.3 (Supplementary Table 1). Consistent with these observa-
tions, SEC, and negative-stain EM data of CtBP1(K46W) with these RAI2
variants showed increased retention volume, indicating a significant
reduction of ordered filament formation (Supplementary Fig. 14,
Supplementary Table 2). Taken together, our data demonstrate that

ordered CtBP1/RAI2 filament formation depends on the presence
of RAI2 for proper interface formation of adjacent CtBP1 tetramer
layers.

## RAI2-induced polymerization relieves CtBP repressor activity
To study the effects of RAI2-induced CtBP polymerization in a relevant
tumor cell model, we chose the VCaP PC cell line because of its high
endogenous RAI2 expression and hence the presence of distinct CtBP/
RAI2 foci (Fig. 4a, b, Supplementary Figs. 15, 16a, b). In contrast, no
such foci were observed in VCaP cells lacking RAI (VCaP KO) (Fig. 4b,
Supplementary Figs. 15, 16b). To investigate the effect of RAI2-
mediated CtBP polymerization on the transcriptional regulation of
the tumor suppressor gene *CDKN1A*[9,21], we first performed an in vitro
transactivation assay in human embryonic kidney (HEK) 293T cells. We
found that co-expression of RAI2 and CtBP1 resulted in a 2.5-fold
increase in *CDKN1A* promoter activity, suggesting relief of CtBP cor-
epressor activity in the presence of RAI2 (Supplementary Fig. 17).
Consistent with this, levels of the *CDKN1A* encoded protein p21
(CDKN1A) were low in VCaP KO cells but significantly increased in
parental (PAR) VCaP cells with concomitant RAI2 expression (Fig. 4d).
Under genotoxic stress conditions using hydrogen peroxide ($H_2O_2$) or
Camptothecin (CPT) treatment[22], both *CDKN1A* gene expression and
CDKN1A protein levels were further increased in VCaP PAR cells
(Fig. 4c, d), accompanied by increased number and volume of CtBP/
RAI2 foci (Supplementary Figs. 15, 16c). In VCaP KO cells, we did not
observe any of these effects, demonstrating that RAI2 relieves the
repression of *CDKN1A* (Fig. 4c, d).

## RAI2-induced CtBP polymerization associates with the inhibition of EZH2 activity
Since CtBP repressor function is linked to PRC2 activity[3], we next
examined the effects of CtBP in the absence and presence of RAI2 on
key PRC2 markers, including Enhancer of zeste homolog 2 (EZH2) and
the EZH2-induced post-translational modification H3K27me3, by
chromatin immune precipitation (ChIP) (Fig. 4e). As expected, we
found impaired binding of CtBP to the *CDKN1A* promoter in VCaP PAR
cells compared with VCaP KO cells. While the level of EZH2 binding to
chromatin remained unchanged regardless of the presence of RAI2, we
found a strong increase of H3K27me3 at the target promoter in VCaP
KO cells, indicating that EZH2 catalytic activity is in-line with CtBP-
mediated repression of the *CDKN1A* gene (Fig. 4d). Since the level of
histone H3K27me3 modification is low in VCaP PAR cells, our data
suggest that RAI2-mediated CtBP polymerization downregulates EZH2
activity. These data are supported by the colocalization of both EZH2
and H3K27me3 with CtBP/RAI2 foci in VCaP PAR cells (Fig. 4f, g). In
contrast, in the absence of CtBP1/RAI2 foci we found only a diffuse

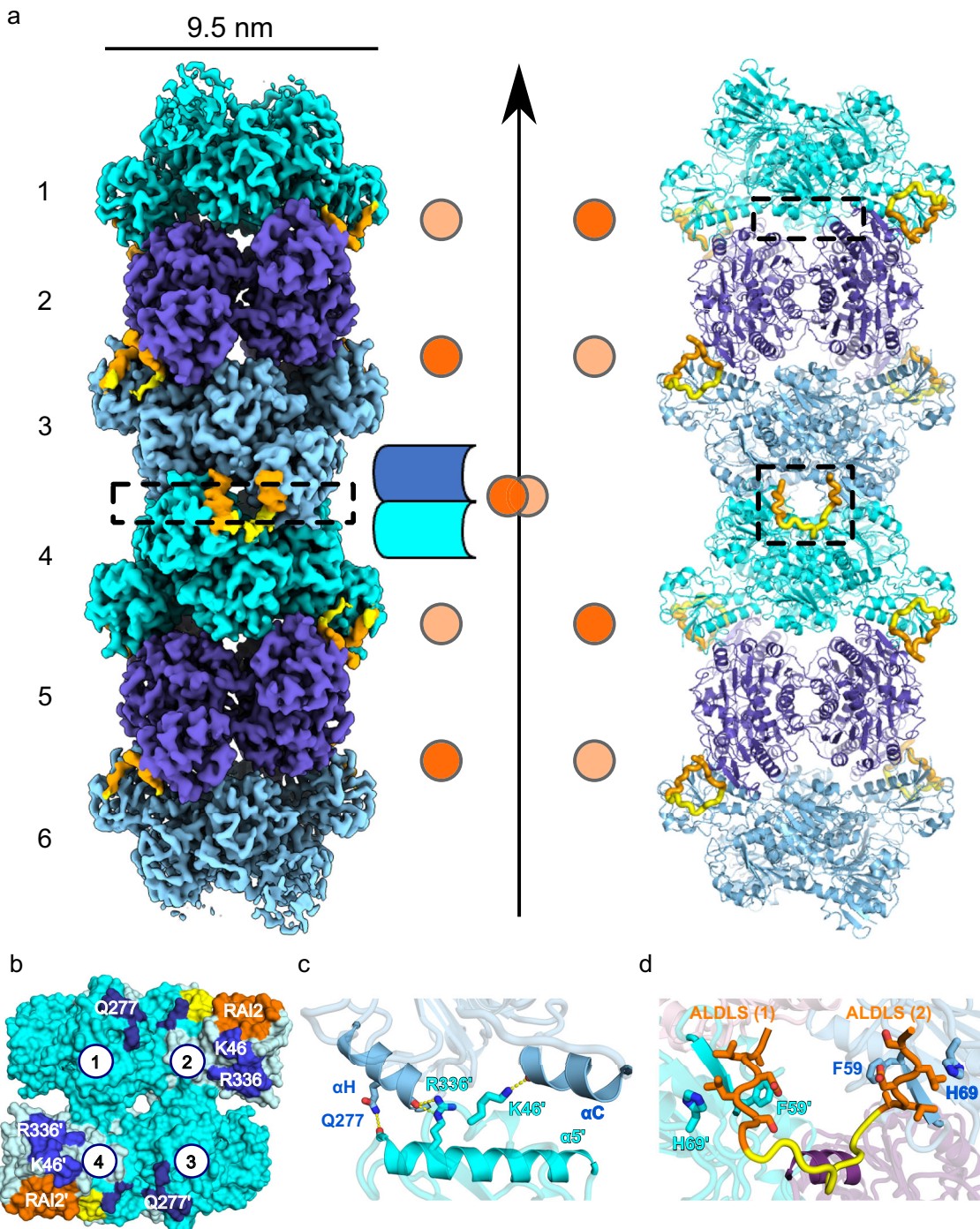

**Fig. 3 | Molecular basis of RAI2-induced CtBP polymerization. a** Cryo-EM density (left), schematic (middle), and cartoon representation of the CtBP1(WT, FL)/RAI2(Δ11, 303–362) filament confined to six CtBP1 tetramer layers (right) that are colored in cyan, violet, and light blue. Adjacent CtBP1 layers are rotated around a vertical 3-fold filament axis and are connected by a pair of RAI2 peptides (yellow) with an ALDLS tandem motif (orange) in opposite positions to the central filament axis. Book-like opening of a CtBP1(n)/CtBP1(n + 1) interface (boxed) without bound RAI2 is indicated schematically. **b** Top view of the RAI2-mediated CtBP1(n)/CtBP1(n + 1) interface in cyan and pale cyan (boxed in **a**, left). Areas interacting with the next tetrameric CtBP layer are in blue and dark blue. The four CtBP subunits are numbered as in Fig. 2e. Residues involved in specific CtBP1/CtBP1 interface interactions are indicated (*cf.* **c**). RAI2-interacting surface segments are in orange (ALDLS motif) and yellow (other residue segments). **c** Side view of the CtBP1(n)/CtBP1(n + 1) interface (boxed in **a**, left), highlighting secondary structural elements and residues that contribute to specific interface interactions (*cf*. Supplementary Fig. 13). **d** Side view of the visible RAI2 segment interacting with two adjacent CtBP layers (boxed in **a**, right). The connecting RAI2 tandem ALDLS linker (yellow) is in close contact with a third CtBP subunit (violet). See also Supplementary Figs. 10–14 and Supplementary Tables 3 and 4.

distribution of these components within the nuclei of VCaP KO cells (Supplementary Fig. 18a, b). CtBP2 showed the same pattern of colocalization with EZH2 and H3K27me3 as CtBP1 (Supplementary Fig. 18c–f). Interestingly, these findings are in agreement with a highly significant anti-correlation of gene expression levels of *RAI2* and *EZH2* in a primary prostate adenocarcinoma cohort, indicating that both genes may have opposing effects in primary prostate tumors[23] (Supplementary Fig. 19).

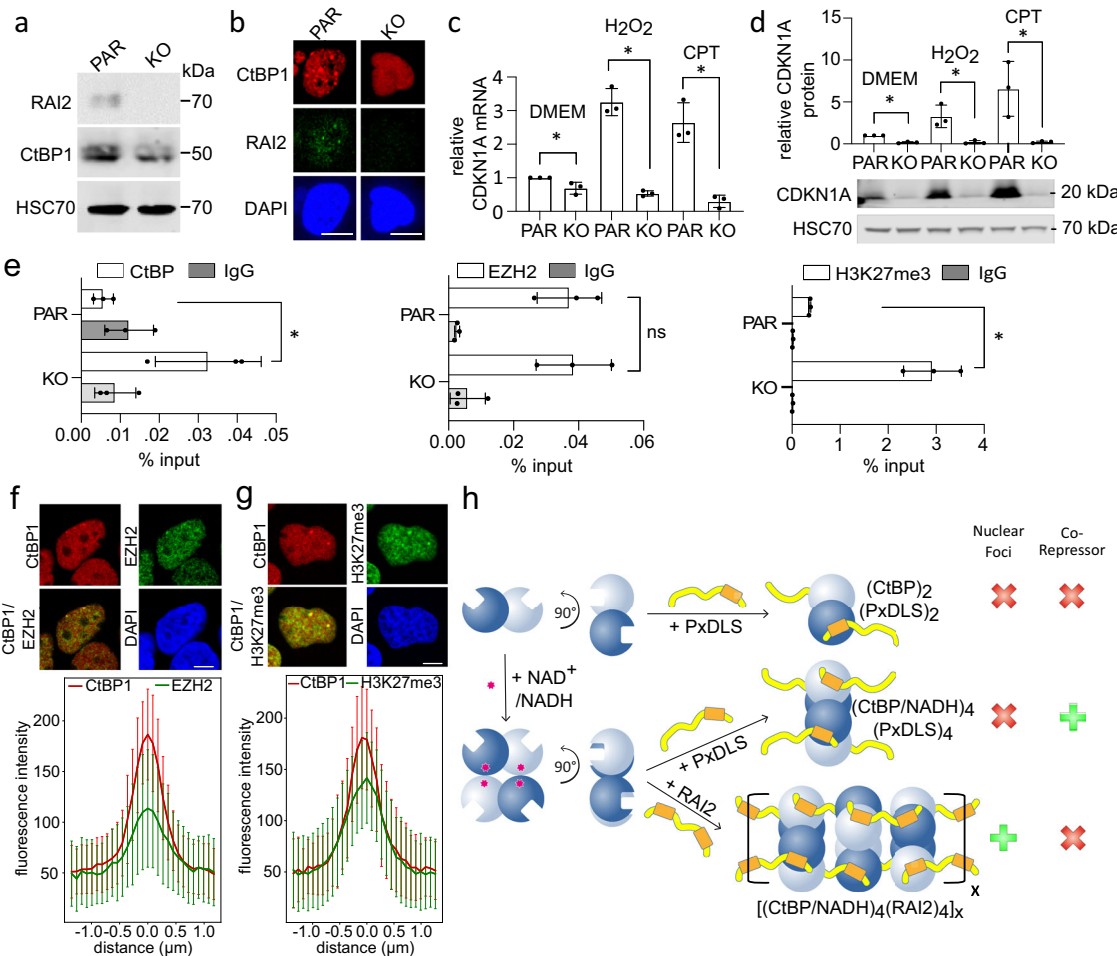

**Fig. 4 | RAI2-induced CtBP polymerization relieves the *CDKN1A* gene repression by downregulating EZH2 activity. a** Expression levels of RAI2, CtBP1 and HSC70 in VCaP PAR and KO cells. **b** representative immunofluorescence staining of CtBP1 (red) RAI2 (green). DAPI (blue) staining was used for localization of nuclei. Scale bars: 10 μm. **c, d** *CDKN1A* gene (**c**) and CDKN1A protein (**d**) expression levels in VCaP PAR and KO cells, untreated (DMEM), $H_2O_2$ and CPT. Data are represented as the mean of n = 3 independent biological experiments ± SD. Unpaired t-test was used to evaluate statistical significance applying p values for a two-sided confidence interval as follows: p < 0.001 (***), p < 0.01 (**), p < 0.05 (*). Definition p values: qPCR: p = 0.04 DMEM, p < 0.001 $H_2O_2$, p < 0.001 CPT; Western Blot: p < 0.001 DMEM, p = 0.017 $H_2O_2$, p = 0.029 CPT. **e** ChIP-qPCR analysis of CtBPs, EZH2. H3K27me3 and IgG (negative control) binding to *CDKN1A* promoter in VCaP PAR and KO cells. The data are represented as mean of n = 3 independent biological experiments ± SD. Unpaired t-test was used to evaluate statistical significance applying p values for a two-sided confidence interval as follows: p < 0.001 (***), p < 0.01 (**), p < 0.05 (*), "ns" not significant change. Definition p values: p = 0.028

CtBP, p = ns EZH2, p = 0.002 H3K27me3. **f, g** Colocalization of CtBP1 (red) with EZH2 (green) or H3K27me3 (green) in VCaP PAR cell nuclei. Mean fluorescence intensity profiles ± SD of CtBP1 overlayed with EZH2 or H3K27me3 images (bottom panels) by line scanning of 100 foci each over 3 μm distance. 100 foci from one representative experiment were analyzed for each condition. Colors are as in the images used for analysis. Scale bars: 10 μm. **h** Model of CtBP of regulation of co-epressor activity by RAI2-induced polymerization. While CtBP dimerization is NAD⁺/NADH-independent, CtBP tetramerization is mediated by redox-sensitive NAD⁺/NADH binding (indicated by asterisk). Binding of PxDLS SLiM-containing proteins does not require a specific CtBP assembly state and results in CtBP/PxDLS target complexes with multiple 1:1 stoichiometries. Tandem ALDLS motif-containing RAI2 induces CtBP polymerization into CtBP/RAI2 filaments with an overall [4:2]ₓ stoichiometry, which leads to nuclear CtBP/RAI2 foci and inactivates CtBP as a master PRC2 corepressor. See also Supplementary Figs. 15–19. Source data are provided as a Source Data file.

## Relevance of RAI2 gene expression in prostate cancer progression

To demonstrate the potential clinical relevance of RAI2-induced CtBP polymerization, we probed its role in the stepwise progression of PC[5,24]. Under androgen deprivation therapy, PC often progresses from the hormone-sensitive (HSPC) to castration-resistant (CRPC) stage and may further transdifferentiate to treatment-resistant variants including tumor lesions with neuroendocrine traits[24,25]. Investigating the antagonistic roles of CtBP and RAI2 in PC progression is relevant because both proteins have been shown to be independently involved in PC-associated androgen signaling processes[26–28]. To this end, we performed gene expression analyses of circulating tumor cells (CTCs) enriched from blood samples of metastatic PC patients from HSPC, CRPC, aggressive variant PC (AVPC), and histologically evident

neuroendocrine prostate cancer (NEPC)[29,30]. From more than 100 blood samples analyzed (Supplementary Data 7), the percentage of patients with detectable CTCs increased from 67 to 70% in HSPC/CRPC patients to 89-91% in treatment-resistant AVPC/NEPC disease stages (Fig. 5a). Notably, while *RAI2* gene expression was detected in 89% of CTC-positive blood samples in CRPC patients, this rate decreased to 35% in AVPC and 19% in NEPC patients (Fig. 5b), suggesting that loss of RAI2 to be associated with PC progression to treatment-resistant disease stages. Since EZH2 is a known driver of CRPC development and transdifferentiation to NEPC[31,32], we tested whether RAI2 loss alone could induce neuroendocrine traits. Indeed, we detected a strong increase in the expression levels of the neuroendocrine markers synaptophysin (*SYP*) and chromogranin (*CHGA*) in VCaP KO cells when compared to VCaP PAR cells (Fig. 5c, d). We also observed a higher

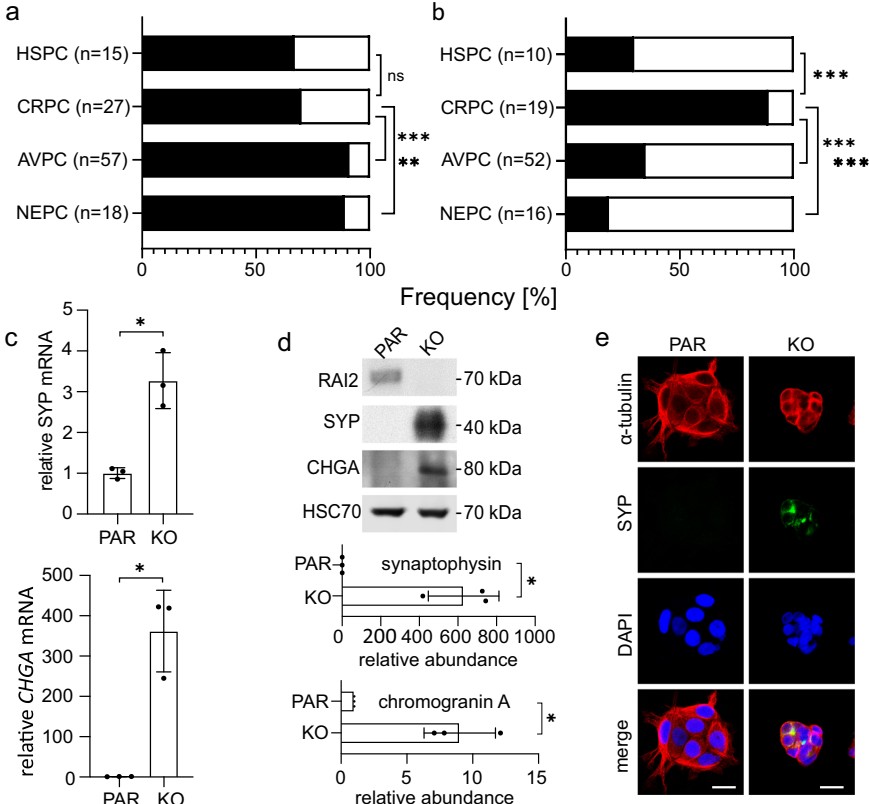

**Fig. 5 | Effect of RAI2 loss on PC progression. a** Frequency (black bars) of CTC detection in blood samples from HSPC (n = 15), CRPC (n = 27), AVPC (n = 57) and NEPC (n = 18) patients. Fischer's exact test was used to calculate the p values with confidence values as follows: p < 0.001 (***), p < 0.01 (**), p < 0.05 (*), "ns" = not significant change. The CRPC cohort was used as the reference. Definition p values: ns HSPC, p < 0.001 AVPC, p = 0.001 NEPC. **b** Frequency (black bars) of *RAI2* mRNA detection in CTC-positive blood samples (n = 10 for HSPC, n = 19 for CRPC, n = 52 for AVPC, n = 16 for NEPC). Fischer's exact test was used to calculate the p values applying the confidence values as follows: p < 0.001 (***), p < 0.01 (**), p < 0.05 (*). The CRPC cohort was used as the reference. Definition p values: p < 0.001 for all three cohorts. **c** *SYP* and *CHGA* gene expression levels in VCaP PAR and KO cells. Data are represented as the mean of n = 3 independent biological experiments ± SD. Unpaired t-test was used to evaluate statistical significance with p values for a

two-sided confidence interval as follows: p < 0.001 (***), p < 0.01 (**), p < 0.05 (*). Definition p values: p = 0.005 SYP, p = 0.004 CHGA. **d** Upper panel, protein expression levels of RAI2, synaptophysin (SYP), chromogranin A (CHGA), and HSC70 (loading control) in VCaP PAR and KO whole cell extracts. Bottom panel, densitometric quantification. The data are represented as mean of three independent biological experiments ± SD. Unpaired t-test was used to evaluate statistical significance applying p values for a two-sided confidence interval as follows: p < 0.001 (***), p < 0.01 (**), p < 0.05 (*). Definition p values: p = 0.005 SYP, p = 0.004 CHGA. **e** Representative immunofluorescence staining of VCaP PAR and KO cells stained for α-tubulin (red), synaptophysin (green), DAPI (blue), and merge of α-tubulin DNA (red-blue). Scale bars: 25 μm. See also Supplementary Data 7. Source data are provided as a Source Data file.

nucleus/cytoplasm ratio and smaller cell size in VCaP KO cells compared to VCaP PAR cells, which is also indicative of NEPC[30] (Fig. 5e). In conclusion, our data show that loss of RAI2 in prostate tumor cells matches our observations in the development of treatment-resistant neuroendocrine traits, in agreement with our molecular data demonstrating that RAI2-induced CtBP polymerization is required for the relief of transcriptional repression.

## Discussion

Unraveling the principles and mechanisms of regulation of biological activities remains a challenging topic in the life sciences. Those involving mostly folded proteins such as allosteric processes, changes in assembly state and post-translational modifications are well established, but insight into molecular mechanisms involving unfolded proteins is more challenging and requires innovative approaches with biophysical methods, to provide structural insight at low resolution. This is particularly true for processes involving SLiMs of unfolded proteins[33,34]. Here, we have shown that the RAI2 protein without a folded domain structure has a unique ability to induce CtBP polymerization via multivalent interaction sites from both binding partners. The twofold repeated RAI2 ALDLS motif belongs to a larger group of SLiMs with a PxDLS consensus sequence that occur in one or more

copies in a variety of CtBP-binding proteins[16,35]. Our data thus add an additional dimension to the regulation of CtBP activity through polymerization mediated by multiple SLiMs (Fig. 4h). This inactivation mechanism complements the demonstrated loss of activity in the absence of NAD⁺/NADH, which is required for CtBP tetramer assembly and induction of its metabolic- and redox-sensing properties[36–39]. Since CtBP binding of PxDLS protein targets does not require oligomerization[16,40], CtBP tetramerization has been proposed as a prerequisite for simultaneous recruitment of multiple chromatin modifying enzymes and/or proper nuclear localization[13,41]. In this model, CtBP polymerization would counteract tetrameric CtBP as the functionally active assembly.

There is only a limited understanding of protein polymerization mediated by multivalent interactions, especially involving intrinsically disordered SLiM motif-containing protein partners that match the overall structural features of RAI2[42,43]. Available data are suggestive of models proposing molecular condensates that could generate membraneless compartments with enhanced local concentrations of target proteins[44–46]. Interestingly, both PRC1 and PRC2 machineries are known to generate phase-separated condensates known as "Polycomb bodies" for efficient chromatin compaction, induced by interactions with disordered protein components[47]. Our observation that EZH2

colocalizes with CtBP/RAI2 foci (Fig. 4f) suggests that RAI2-mediated CtBP polymerization may have implications for PRC2 compartmentalization.

Due to its multifunctionality, well-defined domain organization and the requirement for oligomeric assembly to act as a major co-repressor, CtBP offers multiple opportunities for therapeutic intervention[48,49]. Our data on RAI2-mediated CtBP polymerization, which presents a powerful mechanism of CtBP repressor inactivation may inspire currently untested routes for drug development, by chemically exploiting the recurrence of two PxDLS motifs. Such polymerization could potentially interfere with crucial transition processes leading to increased cancer malignancy, as demonstrated for the CRPC/NEPC transition based on our findings in this contribution and EMT-like processes in other cancer types. The extent to which the molecular principle of target deactivation by polymerization through multivalent interactions is applicable to other molecular processes under healthy and diseased conditions provides an intriguing future perspective.

## Methods

### Ethics
Human blood samples were collected at the Center for Oncology, University Medical Center Hamburg-Eppendorf between June 2018 and July 2022 in accordance with the Declaration of Helsinki and approved by the Ethics Committee of the City of Hamburg (Hamburger Ärztekammer, PV5392), and all patients provided informed consent. No compensation was provided to donors in this study. All samples were anonymized prior to processing.

### Secondary and tertiary structure prediction
IUPRED2A was employed for sequence disorder prediction[50,51]. AlphaFold/ColabFold was used for tertiary structure prediction[52–54].

### Plasmid construction
Plasmids pcDNA3.1-hCtBP1 isoform 1 (Uniprot ID: Q13363) and pCMV-SPORT6-CtBP2 (Uniprot ID: P56545) were purchased from Genescript USA Inc (Piscataway, NJ, USA) and Addgene (Watertown, MA, USA), respectively. CtBP1 and CtBP2 were cloned into pETM14 (EMBL, https://grp-pepcore.embl-community.io/vectors/ecoli.html, Germany) using restriction sites NcoI and XhoI restriction sites in frame with an N-terminal polyhistidine tag and a 3C protease site sequence LEVLFGP. The PCR primers used were ATGGGCAGCTCGCACTTG and CAACTGGTCACTGGCGTGG for CtBP1 and ATGGCCCTTGTGGATA AGCA and TTGCTCGTTGGGGTGCTCTC for CtBP2. CtBP1 and CtBP2 mutants were generated using Plasmids for RAI2 protein expression in mammalian cells were constructed as described[17]. The cDNA of truncated versions of RAI2 were purchased as gene blocks (Integrated DNA Technologies, IA, USA) and were cloned into the expression vector pETGB1a (EMBL), using the NcoI and XhoI restriction sites in frame with an N-terminal hexahistidine G-binding protein domain fusion tag and a tobacco etch virus (TEV) protease site with sequence ENLYFQG. RAI2 variants M1 (L319A, S320A), M2 (L345A, S346A), and M1+M2 (L319A, S320A, L345A, S346A) were also generated using gene blocks (Integrated DNA Technologies, IA, USA) and cloned as mentioned above. All vectors were confirmed by DNA sequencing.

### Protein expression and purification
CtBP1 and CtBP2 variants were expressed in *Escherichia (E.) coli* strain BL21(DE3) pLysS. Cultures were grown in the Terrific Broth medium at 37 °C to an optical density of 1.2, induced with 0.2 mM isopropyl thiogalactose (IPTG) for 18 h, pelleted by centrifugation (30 min, $5000 \times g$) and resuspended in 50 ml lysis buffer containing 50 mM HEPES/NaOH (pH 7.5), 300 mM NaCl, 10 mM imidazole, 10% (v/v) glycerol. Cells were lysed three times by using an emulsifier at a constant pressure of 10,000 psi. Cell debris was removed by centrifugation

(45 min, $21,000 \times g$), passed through a 0.45 μM filter, incubated with nickel-charged affinity (Ni-NTA) beads in 10 ml slurry and equilibrated with lysis buffer for 60 and passed through it. The column was then washed with 100 ml lysis buffer, followed by 100 ml wash with lysis buffer containing 30 mM imidazole and 1 M NaCl and finally eluted with a 30 ml volume of 200 mM imidazole. The eluted fusion proteins were cleaved with 3C protease in overnight dialysis buffer, which was identical to the lysis buffer. The cleaved CtBP proteins were again passed through Ni-NTA beads equilibrated with lysis buffer, to remove 3C protease and further impurities. Proteins were dialyzed in 4 liters of 50 mM Tris/HCl (pH 8.0), 5% [v/v] glycerol for 6 h and then passed through a pre-equilibrated HiTrap Q HP 5 ml column (Cytiva, MA, US) and eluted with 125 ml of a 0–1,0 M NaCl gradient to yield pure proteins as confirmed by sodium dodecyl sulfate (SDS)-polyacrylamide gel electrophoresis (PAGE) analysis. Highest purity protein fractions were pooled, dialyzed against SEC buffer (50 mM HEPES (pH 7.2), 150 mM NaCl), concentrated, and loaded onto a Superdex 200 column (Cytiva, MA, USA). Due to the presence of disordered N- and C-termini, CtBP1(28–353) and CtBP2(31–364) were used for NMR spectroscopy and X-ray crystallography experiments, respectively.

RAI2(303–362), RAI2(303–465) and RAI2(303–530) variants were transformed into BL21 DE3 LOBSTR *E. coli* cells and plated on kanamycin plates. A single colony from the plates was inoculated overnight into 100 ml of Luria-Bertani (LB) medium containing 50 μg/ml kanamycin. This primary culture was inoculated into 3 liters of Terrific Broth medium to an $OD_{600nm}$ of 1.2, protein expression induced with 0.5 mM IPTG for 18 h and pelleted by centrifugation (30 min, $5000 \times g$). For NMR spectroscopy, primary cultures were inoculated into 800 ml LB medium until an $OD_{600nm}$ of 0.7, centrifugated at $6000 \times g$ for 20 and then inoculated into 2 liters of M9 medium supplemented with 3 g/l $^{15}NH_4Cl$ (1.5 g/l) and $^{13}C_6$-glucose. Protein expression was induced at an $OD_{600nm}$ of 0.7 by the addition of 0.5 mM IPTG at 18 °C for 18 h and then pelleted by centrifugation (30, $5000 \times g$).

Pellets were resuspended in 50 ml lysis buffer containing 50 mM HEPES/NaOH (pH 7.5), 300 mM NaCl. Cells were lysed three times by using an emulsifier at a constant pressure of 10,000 psi and cell debris was removed by centrifugation (40, $21,000 \times g$), passed through a 0.45 μM filter and loaded onto a pre-equilibrated HisTrap HP 5 ml column (GE Healthcare, IL, US). The column was washed with 50 ml lysis buffer and eluted with buffer containing 200 mM imidazole. The eluted fusion proteins were cleaved with TEV protease in the overnight dialysis buffer. The cleaved proteins were again passed through the same HisTrap HP 5 ml column equilibrated with lysis buffer to remove TEV protease and fusion tags. The flowthrough was then concentrated using a 3.5 kDa concentrator (Merck KgaA, Germany) and loaded onto a Superdex 75 column SEC column (Cytiva, MA, USA) pre-equilibrated with 50 mM HEPES (pH 7.2), 150 mM NaCl buffer. Each purification step was analyzed by SDS-PAGE before proceeding to the next step. Purified proteins were concentrated, aliquoted, and stored at −80 °C or used for further biophysical and high-resolution structural biology experiments.

### Mass spectrometry
Coomassie-stained bands of RAI2(WT, 303–530) were excised, cut into small pieces, and transferred to 0.5 ml Eppendorf tubes. For all subsequent steps, buffers were exchanged by two consecutive 15 min incubation steps of the gel pieces with 200 μl acetonitrile (ACN), with ACN removed after each step. Proteins were reduced by adding 200 μl of 10 mM dithiothreitol (Biomol, Hamburg, Germany) solution in 100 mM ammonium bicarbonate (AmBiC, A6141, Sigma-Aldrich, MA, USA) and incubating at 56 °C for 20 min. 180 μl of ACN was added after 15 min of incubation at room temperature (RT). Proteins were alkylated by adding 200 μl of a 55 mM chloroacetamide solution in 100 mM AmBiC. Samples were incubated for 20 min in the dark. Gel pieces were

incubated twice with 200 μl ACN for 15 min at RT. Dried gel pieces were transferred to glass vials (Chromacol glass inserts; #8871160101226, ThermoFisher Scientific, MA, USA) and placed into a 2 ml Eppendorf cup filled with 700 μl water. Subsequently, 50 μl of 3 M HCl was added and the gel pieces were incubated for 5 min at RT. Samples were then transferred into a microwave where they were heated for 10 min at 1000 W. Samples were spun down and the supernatant was directly subjected to a reverse-phase cleanup step (OASIS). Peptides were dried in a speedvac and reconstituted in 10 μl of an aqueous solution of 0.1% (v/v) formic acid. Peptides were analyzed by liquid chromatography–tandem mass spectrometry (LC-MS/MS) on an Orbitrap Fusion Lumos mass spectrometer (ThermoFisher Scientific, MA, USA)[55]. Peptides were separated using an Ultimate 3000 nano rapid separation liquid chromatography system (Dionex, Thermo-Fisher Scientific, MA, USA) equipped with a trapping cartridge (Pre-column C18 PepMap 100, 5 mm, ThermoFisher Scientific, MA, USA) and an analytical column (Acclaim PepMap 100. 75 × 50 cm C18, 3 mm, 100 Å, ThermoFisher Scientific, MA, USA), connected to a nanospray-Flex ion source (ThermoFisher Scientific, MA, USA). Peptides were loaded onto the trap column with solvent A (0.1% formic acid) at 30 μl per min and eluted with a gradient of 2 to 85% solvent B (0.1% formic acid in acetonitrile) over 30 min at a flow rate of 0.3 μl per min. All solvents were of LC-MS grade. The Orbitrap Fusion Lumos mass spectrometer was operated in positive ion mode with a spray voltage of 2.2 kV and a capillary temperature of 275 °C. Full scan MS spectra with a mass range of 350–1500 m/z were acquired in profile mode with a resolution of 120,000 (maximum injection time of 100 ms). The automatic gain control target was set to standard, and the RF lens setting was kept at 30%. Precursors were isolated using the quadrupole with a 1.2 m/z window and fragmentation was triggered by higher energy collisional dissociation in fixed collision energy mode with a fixed collision energy of 30%. MS spectra were acquired in ion trap normal mode with dynamic exclusion was set to 5 s. The acquired data were processed using IsobarQuant[56] and Mascot (v2.2.07) using a reversed Uniprot E. *coli* database (UP000000625) that included common contaminants. The following modifications were considered: Carbamidomethyl (C) (fixed modification), Acetyl (N-terminal) and Oxidation (M) (variable modifications). The mass error tolerance was set to 10 ppm for full scan MS spectra and to 0.02 Da for MS/MS spectra. A maximum of 2 missed cleavages were allowed. A minimum of 2 unique peptides with a peptide length of at least 7 amino acids and a false discovery rate below 0.01 were required at the peptide and protein level. The result obtained from the run is shown in Supplementary Data 8.

## Quantitative determination of binding affinities

Isothermal titration calorimetry (ITC) experiments were carried out with a VP-ITC system (Malvern Pananalytical, UK). Experiments were performed at 25 °C in 50 mM HEPES (pH 7.2), 150 mM NaCl. Proteins were dialyzed overnight against this buffer. Purified CtBP1 and CtBP2 were placed in the reaction cell while the RAI2 constructs were loaded into the injection syringe. The CtBP concentration used was 10 μM for RAI2(M1, 303–362) or RAI2(M2, 303–362) in the syringe or 20 μM for the WT RAI2 constructs. The concentration of the RAI2(303–362) variants in the syringe was 100 μM. Injections of 10 μl of RAI2(303–362) variants were performed at 4-min intervals. ITC data were processed using NITPIC, SEDPHAT and GUSSI[57]. Values reported are the mean of three independent measurements, and the ± errors represent SDs. Significance between CtBP1(WT)/RAI2(WT, 303–362) and other CtBP/RAI2 complexes was evaluated by two-sided Student's t-test and corresponding p values are shown.

## Analytical size exclusion chromatography

Complex formation between CtBP1 and CtBP2 variants and RAI2 variants was investigated on an analytical Superose 6 3.2/300 GL column

(Cytiva, MA, USA) coupled to a 1260 Infinity HPLC system (Agilent, CA, USA). Proteins were dialyzed against the SEC buffer and injected onto a pre-equilibrated S6 column using an autosampler. The concentration of CtBP1 and CtBP2 variants was kept constant at 100 μM and RAI2 variants were titrated at stoichiometric ratios of 1:0.1, 1:0.2, 1:0.5, 1:1 and 1:2. The gel filtration protein standard containing thyroglobulin (670 kDa), γ-globulin (158 kDa), ovalbumin (44 kDa), myoglobin (17.5 kDa) and vitamin B12 (1.35 kDa) was used to calibrate the column (Bio-Rad, CA, USA). Curves were plotted using GraphPad Prism (Dotmatics, MA, USA).

## Circular dichroism (CD) spectropolarimetry

Prior to each measurement, protein samples were dialyzed against 10 mM $K_2PO_4$ (pH 7.2), 100 mM NaF, and diluted to a concentration of 0.25 mg/ml. Spectra were recorded at 20 °C on a Chirascan CD spectrometer (Applied Photophysics, UK), between 180 and 260 nm in a 0.1 cm cuvette. The number of scans per measurement was 10. The instruments settings were as follows: 1 nm bandwidth, 0.5 s response time, and 1 nm data spacing. Spectra were background subtracted, and converted to mean residue ellipticity [Θ] using Chirascan (Applied Photophysics, UK) and the curves were plotted using GraphPad Prism (Dotmatics, MA, USA). The secondary structure content of the proteins was calculated using the CDNN neural network[58].

## Small Angle X-ray Scattering (SAXS)

SAXS data were collected on beamline P12 at PETRA III (EMBL/DESY, Hamburg, Germany), using a PILATUS 6 M pixel detector (DECTRIS, Switzerland). A total of 40 μl sample volume was exposed to X-rays while flowing through a temperature controlled 1.2 mm wide quartz capillary at 20 °C. A total of 40 image frames of 0.2 s exposure time each were collected. Data were normalized to the transmitted X-ray beam intensity, averaged, subtracted the buffer contribution, and placed on absolute scale relative to water using SASFLOW[59,60]. Data correction was performed using PRIMUS*qt* and the ATSAS 3.0.1 software package[61]. Forward scattering intensity $I(0)$ and radius of gyration $R_g$ were determined from Guinier analysis assuming that at angles $s \leq 1.3/R_g$ the intensity is represented as $I(s) = I(0)\exp(-(sR_g)2/3)$. These parameters were also estimated from the full SAXS curves using the indirect Fourier transform method implemented in GNOM[62], along with the distance distribution function $p(r)$ and estimates of the maximum particle dimensions $D_{max}$. Molecular masses of solutes were estimated from $I(0)$ by calculating partial specific volume and contrast between the protein sequence and buffer components. Theoretical scattering intensities were computed from structural coordinates with CRYSOL[63].

Low resolution shapes were reconstructed from SAXS data using DAMMIF[64]. Rigid body models for different CtBP1/RAI2(303-465) and CtBP1/RAI2(303-362) complexes were computed from the experimental data using CORAL[60] followed by the generation of fits against experimental scattering data in solution using CRYSOL. The available high-resolution crystal structures of CtBP1 (PDB ID: 6CDF) and CtBP2(WT, 31-364)/RAI2(315-321) were used as input rigid bodies. For the full-length CtBP1 tetramer, a monomer was extracted from 6CDF and P222 symmetry was applied during structure calculation. Additional residues corresponding to the 24 and 83 N-terminal and C-terminal intrinsically disordered residues missing from the crystal structure, respectively, were modeled as dummy residues. This tetramer along with the interface information obtained from our crystal structure was then further used to calculate the stoichiometric 4:4 complexes for the RAI2 M1, M2, and M1+M2 mutants. For the large polymeric WT complex formed between CtBP1 and RAI2, tetrameric CtBP1 assemblies were used as rigid bodies and tetramers linked by RAI2 monomers at CtBP1 binding sites identified by consensus between the crystallographic data and NMR chemical shift analysis. For the polymeric Δ11 complex, a theoretical scattering plot was generated

for the cryo-EM structure and the fits were generated against the WT and Δ11 experimental scattering data. The goodness of the fits is reflected in the respective χ² values. Kratky plots were obtained using ATSAS.

## X-ray structure determination

Human CtBP2(31–364)/RAI2(M2, 303–465) complex at a concentration of 10 mg/ml was crystallized from 20% [w/v] PEG-6000, 100 mM Tris (pH 7.5) at 4 °C. Prior to X-ray data collection, the crystals were cryoprotected with 25% [v/v] glycerol, mounted on cryo-loops (Hampton Research, CA, USA) and were flash frozen in liquid nitrogen. X-ray diffraction data were collected at beamline P13 at PETRA III (EMBL/DESY, Hamburg, Germany). The data were integrated using XDS[65], scaled, and merged using AIMLESS[66] within the CCP4 suite[67]. The structure was determined by molecular replacement using Phaser[68], using another CtBP2 structure (PDB ID: 2OME) as a search model. The asymmetric unit of these crystals contained four CtBP2 molecules. The fourth molecule was disordered due to poor crystal packing, as evidenced by the high B-factors of the final structural model. After partial coordinate refinement, the *2Fo-Fc* map resolved distinct electron density for three RAI2 peptides with side-chain features that enabled the modeling of seven residues 315–321 (sequence: EALDLSM) of each RAI2 polypeptide chain. Iterative model building and refinement were performed in COOT (v0.8.9)[69,70], Phenix (v1.13)[71], and Refmac (v5.8.0267)[72], using non-crystallographic symmetry (NCS) constraints and the translation-libration-screw-rotation parameters. The final model was refined to an R/Rfree of 0.23/0.29 at 2.64 Å resolution. For further structural analysis, a biological tetramer was generated using polypeptide chains A and B and the equivalent chains of a crystallographic symmetry mate. The quality of the model was assessed by Ramachandran Plot analysis with 95% of the residues in the most favored position, 4.6% as allowed, and 0.4% as the outliers.

## NMR spectroscopy

NMR data were acquired at 297 K in 20 mM $Na_2PO_4$ and 50 mM NaCl (pH 6.8), 10% $D_2O$, and 1.5 mM tris(2-carboxyethyl) phosphin hydrochloride on a 600 MHz NMR spectrometer (Bruker, MA, USA) equipped with a cryo-probe and a 700 MHz Avance III spectrometer. 2D [¹H,¹⁵N] heteronuclear single quantum coherence (HSQC) spectra were recorded using States/TPPI quadrature detection mode in the indirect dimension. Typically, 512 × 2048 complex points with 8 transients were used to record the 12–14 and 24–28 ppm spectral widths in the ¹H and ¹⁵N dimensions, resulting in acquisition times of 0.150 × 0.135 s, respectively. The transmitter frequency for the proton channel was placed in resonance with the water signal at a ¹⁵N frequency offset of 119–120 ppm. The ¹⁵N-{¹H} Nuclear Overhauser Effect (NOE) was recorded in echo-antiecho detection mode in the indirect dimension in an interleaved manner with and without proton saturation using 5 ms separated 120° pulse trains for a total of 5 s as recycle delay[73]. Residue-specific heteronuclear NOEs were calculated as the ratio of ¹⁵N–¹H cross-peak intensities ($I_{sat}/I_{unsat}$) using CCPNmr[74]. For NMR backbone and side-chain assignments, [¹H,¹⁵N]-HSQC, [¹H,¹³C]-HSQC, HNCO, HN(CA)CO, HNCACB, CBCA(CO)NH, CC(CO)NH, TOCSY-HSQC (80 ms mixing time) and NOESY-HSQC (120 and 200 ms mixing time) were used. For triple resonance experiments, the ¹⁵N channel transmitter was set to 119 ppm with 26 ppm spectral width. The carbon carrier frequencies were set to 38 ppm for HNCACB or 43 ppm for CC(CO)NH in experiments sensitive to backbone and side-chain 13C spins with 76 or 74 ppm width, respectively, while it was set to 173 or 173.5 ppm in experiments sensitive to carbonyl ¹³C spins only with 13 ppm spectral width. Increments in the indirect dimensions were 72 ×128 for HNCACB and 168 for CC(CO)NH, respectively, with 32 ×14 ppm spectral width, respectively, and 119 × 176 for HNCO and HN(CA)CO for ¹⁵N and ¹³C. Transients were 8, 16, or 32 depending on sample concentration, measurement type, and sensitivity. Traditional or non-uniform 25%

sampling acquisition modes were used for 3D measurements. Non-uniform sampling processing was achieved by the multidimensional decomposition method to fill in missing data points[75,76]. Proton chemical shifts were referenced relative to the water signal, with indirect referencing in the ¹⁵N dimension using a ¹⁵N/¹H frequency ratio value of 0.101329118 according to the Biological Magnetic Resonance Bank (BMRB)[77]. Additionally, the spectra of RAI2(WT, 303–362) and its two variants were assigned. Next, CtBP1(28–353) was titrated at concentration ratios of 1:0.4, 1:0.7, and 1:1 stoichiometry to the RAI2 variants at 100 μM concentration, followed by [¹H,¹⁵N]-HSQC experiments to perform chemical shift perturbation measurements. Chemical shift perturbations were calculated using[78]

$$\Delta\delta = \sqrt{\left(\delta_{ref}(1H) - \delta(1H)\right)^2 + \left(\frac{\delta_{ref}(15N)}{6.5} - \frac{\delta(15N)}{6.5}\right)^2}$$

Data were processed using TOPSPIN v.4.0.6 (Bruker, MA, US) and analyzed with CARA (v1.9.0.b2)[79], CCPNmr (v2.4) Analysis and SPARKY[74,80].

## Negative-stain electron microscopy (EM)

Carbon-coated copper grids were glow-discharged for 30 s at 25 mA using a GloQube Plus Glow Discharge System (Electron Microscopy Sciences, PA, USA). Grids were coated with 4 μl protein, incubated for 30 s, and then blotted off from the side. The grids were then washed with 4 μl of 2% $(NH_4)_2MoO_4$ staining solution, followed by 4 μl staining for 30 s. The stain was blotted off from the side and the grids were air dried. The grids were imaged on a Talos L120C transmission electron microscope equipped with a 4 K Ceta CEMOS camera using TIA 4.1.5 (ThermoFisher Scientific, MA, US).

## Cryogenic-EM

Quantifoil R2/2, Cu 200 mesh perforated carbon grids (Jena Bioscience, Germany) were glow-discharged at 25 mA for 60 s using the glow cube device. Purified CtBP1 (WT) and RAI2 (Δ11, 303–362) at a concentration of 50 μM were mixed at 1:1 stoichiometry in a buffer containing 50 mM HEPES (pH 7.2), 50 mM NaCl. Next, 5 μl of the sample was applied to the glow-discharge grid, incubated for 10 s, blotted for 2 s with a blotting force of −12 at 4 °C and 100% humidity, followed by plunge freezing in a ratio of 63:37 propane:ethane ratio, using Vitrobot Mark IV device (ThermoFisher Scientific, MA, US) set at 100% humidity and 4 °C.

Cryo-EM data were acquired on Titan Krios electron microscope operating at 300 kV and equipped with a K3 direct electron detector (Gatan, CA, USA) using a GIF quantum energy filter with a 20-eV slit width (ThermoFisher Scientific, MA, USA). Films were acquired using EPU software in count mode (ThermoFisher Scientific, MA, USA). Each micrograph was fractionated over 50 frames, with a total dose of 50 e⁻/Å² for 1.8 s. Image processing was performed using Cryosparc[81]. Movies were drift-corrected using patch motion correction. The contrast transfer function was performed using patch estimation, and micrographs with a resolution greater than 5 Å were selected for subsequent steps.

Manual picking was performed on a subset of micrographs and picked particles were subjected to 2D classification. Selected classes were used for template-based particle picking on the entire dataset. After three rounds of 2D classification, particles with different orientations were selected to generate an initial model using ab initio reconstruction in Cryosparc[81]. The initial model obtained was used as a reference for heterogeneous 3D refinement with three or more classes to determine the level of heterogeneity in the dataset. Selected classes were subjected to both homogeneous and heterogeneous 3D refinement by enabling defocused global CTF refinement. The refined particles were subjected to local motion correction, followed by homogeneous 3D refinement with or without the imposition of

D2 symmetry resulting in final density maps of 3.0 and 3.2 Å resolution, respectively.

The D2 symmetrized cryo-EM map was fitted with six copies of the CtBP1 tetramer crystal structures (PDB code 6CDF) in UCSF Chimera (v1.13)[82]. RAI2 peptides were manually built in COOT (v0.8.9)[69,70] and were subjected to real space refinement in Phenix (v1.19.2)[71] using the secondary structure restraints and NCS constraints. Ramachandran and rotamer outliers were corrected using ISOLDE (v1.1.2)[83], and the model was further refined in Phenix[71]. Validation of the final model and data was carried out using CheckMySequence[84], MolProbity Server[71,85], and tools from the Phenix suite. Interface and solvent-accessible areas of the fiber complex were calculated using PDBePISA[86].

## Cell culture

KPL-1 (RRID:CVCL_2094, DSMZ #ACC-317), VCaP cells (RRID:CVCL_2235, ATCC #RL-2876) and 293 T (RRID:CVCL_0063, ATCC #CRL-3216) were taken from the cell bank of the Institute of Tumor Biology, Hamburg Germany. In August 2018, the correct identity of VCaP and KPL-1 cells was confirmed by short tandem repeat profiling to exclude cross-contamination between cell lines (Multiplexion, Heidelberg, Germany). During cultivation, cell lines were tested monthly for mycoplasma contamination using the Venor®GeM detection kit (Minerva Biolabs, Germany). Cells were grown as monolayers accordingly in Dulbecco's Modified Eagle Medium (DMEM) supplemented with 10% fetal bovine serum and 2 mM L-glutamine at 37 °C in a humidified atmosphere containing 10% $CO_2$.

To generate VCAP RAI2-KO cells, the guide RNA sequence GGCTCAGCTGATCACCACCG was cloned into the pSpCas9(BB)–2A-GFP plasmid[87] and transfected into parental VCaP cells using Lipofectamine 2000 (ThermoFisher Scientific, MA, US) according to the manufacturer's protocol. After 5 days, single GFP-positive cells were isolated by fluorescence-activated cell sorting and clonally expanded. Successful RAI2 inactivation was verified by immunoblot analysis and Sanger sequencing of individual expanded cell clones. A pool of three individually expanded cell clones was used for all further experiments.

## Quantitative real-time reverse transcription polymerase chain reaction (qRT-PCR) analysis

RNA was extracted from cultured cells during exponential growth using the Nucleospin RNA Kit (Macherey-Nagel, Germany). A total of 1000 ng of RNA from each sample was transcribed using the First Strand cDNA Synthesis Kit (ThermoFisher Scientific, MA, USA) and random hexamers. Each qRT-PCR was performed in technical triplicates using SYBR Green (ThermoFisher Scientific, MA, USA) with the following primers: *CDKN1A*: CATGTGGACCTGTCACTGTCTTGTA, GAAGATCAGCCGGCCGTTTG; *RPLPO*: TGAGGTCCTCCTTGGTGAACA, CCCAGCTCTGGAGAAACTGC. Mean crossing point values were used for each gene. Data were analyzed using the comparative CT Method (ΔΔCT)-method with gene expression of *B2M* used for normalization. The significance between different cell lines and the results of three independent experiments were evaluated by a two-sided Student's *t*-test and the corresponding p values are shown.

## Western blot analysis

Whole-cell extracts from cultured cells were prepared by direct lysis and sonication of cells in a 2% SDS sample buffer containing phosphatase and protease inhibitors (Roche, Germany). Cell extracts were separated in denaturing 8% or 15% (used only for CDKN1A) polyacrylamide gels and blotted onto a nitrocellulose membrane. Proteins were detected by incubation with RAI2 (D4W9P, #97857; Cell Signaling Technology, MA, USA), CtBP1 (Clone 3, #612042; BD Bioscience, NJ, USA), CtBP2 (Clone 16, #612044; BD Bioscience, NJ, USA) and CDKN1A (Clone 12D1, #2947; Cell Signaling Technology, MA, USA) antibodies at 1:2000 dilution. Validation of these antibodies is provided by respective manufacturers. HSC70 protein (clone B-6, #sc-7298; Santa

Cruz Biotechnology, CA, USA, dilution 1:10$^6$) was used as a loading control and HRP- (#7074; Cell Signaling Technology, MA, USA) or IRDye® (#925-32210, #926-68071; Li-Cor Biosciences, Germany) coupled secondary antibodies were used for protein detection with either the Curix 60 processor (Agfa HealthCare, Belgium) on Super RX films (Fujifilm, Japan) or the Odyssey® CLx Imaging System (Li-Cor Biosciences, Germany). For signal quantification, the plot lanes function of the FIJI app (RRID:SCR_002285) was used and normalized, using the signal of the protein of interest to the HSC70 signal on the same membrane. The significance between different cell lines and the results of three independent experiments were evaluated by a two-sided Student's t-test supported by the corresponding p values.

## Immunofluorescence staining

Cells were fixed in 4% paraformaldehyde (PFA) in phosphate-buffered saline (PBS) (pH 7.4) for 10 min, washed 3 times with PBS, and permeabilized with 0.2% Triton X-100 in PBS for 10 min. After incubation with 1% bovine serum albumin (BSA) in PBS for 30 min, cells were further incubated with primary antibodies RAI2 (clone D4W9P, #97857, 1:250; Cell Signaling Technology, MA, USA), CtBP1 (clone 3, #612042, 1:250; BD Biosciences, NJ, USA), CtBP2 (clone 16, #612044, 1:250; BD Biosciences, NJ, USA), EZH2 (D2C9, #5246, 1:125, Cell Signaling Technology, MA, USA) or H3K27me3 (C36B11, #9733, 1:250, Cell Signaling Technology, MA, USA) in 1% BSA (w/v) in PBS for 90 min. Antibodies were validated in accordance with the manufacturer's protocols. After three washes with PBS, specific antibody binding was visualized with Alexa Fluor 488 goat anti-rabbit IgG (H+L) (#A-11008) and Alexa Fluor 546 goat anti-mouse IgG (H+L) (#A-11003) (Life Technologies, CA, USA) in 1% BSA in PBS. After three washes with PBS, nuclei were stained with 4′,6-diamidino-2-phenylindole (DAPI) and mounted in Mowiol (Sigma-Aldrich, MO, USA) according to the manufacturer's instructions. Confocal laser scanning microscopy was performed on a Leica TCS SP5 microscope (Leica Microsystems, Germany). Imaris imaging software (Oxford Instruments, UK) was used to identify and measure the number and size of volumes of CtBP1/RAI2 and CtBP2/RAI2 foci. For this purpose, we applied uniform background subtraction and a lower volume cut-off of 0.1 μm$^3$ in statistical analysis by two-sided Student's *t*-test supported by the corresponding p values.

The colocalization of CtBP1/RAI2 and CtBP2/RAI2 foci with EZH2 or H3K27me3 in VCaP PAR cells was verified using fluorescence intensity profiles measured along lines manually drawn on the microscopy images. First, 100 individual foci of each condition were macroscopically identified in single image stacks. The lines were selected to intersect foci observed on channels corresponding to CtBP1 or CtBP2 ignoring any other signal. This procedure resulted in 100 intensity profiles with two channels for each of the four conditions. The profiles were aligned by applying a horizontal shift maximizing cross-correlation function in the CtBP1 or CtBP2 channel with an arbitrarily selected reference profile. Shifts were applied to a second channel of each profile prior to the calculation of mean intensities and SDs, resulting in approximately 90% colocalization efficiency for CtBP/H3K27me3 and 50% colocalization efficiency for CtBP/EZH2.

## Transactivation assay

The reporter plasmid (CDKN1A/WAF1 promoter, Addgene plasmid #16451) was kindly provided by Dr. Bert Vogelstein (Johns Hopkins University, MD, USA)[88]. HEK 293T cells were transiently co-transfected with individual expression plasmids encoding full-length RAI2 and CtBP1 and the reporter plasmid using Lipofectamine 2000 (Thermo-Fisher Scientific, MA, USA). Co-transfection with the pEGFP-N2 plasmid (Takara Bio, CA, USA) served as a control for different transfection efficiencies. The medium was changed after 24 h, and measurements were performed after 48 h. Cells were rinsed with PBS before 20 μl PBS was added for the measurement of green fluorescent protein fluorescence intensity on an Infinite 200 Pro plate reader (TECAN,

Switzerland), using fluorescence excitation at 465 nm and fluorescence emission at 535 nm. Subsequently, luciferase luminescence was determined by adding 10 μl Luc-pairTM Firefly Luciferase HT Assay Kit solution (Gene Copoeia, MD, USA) in an Infinite 200 Pro plate reader (TECAN, Switzerland). The significance between different cell lines and the results of three independent experiments were evaluated by a two-sided Student's *t*-test and the corresponding p values are depicted.

## Chromatin immunoprecipitation (IP)

Cells were cross-linked with 1% PFA for 10 min at room temperature (RT), washed twice with ice-cold 1× PBS, and scraped in low-salt IP buffer (150 mM NaCl, 50 mM Tris/HCl (pH 7.5), 5 mM EDTA, 0.5% Igepal, 1% Triton X-100, 1× proteinase inhibitor cocktail (Roche, Germany)). After centrifugation, the nuclear pellet was resuspended in 300 μl IP buffer and sonicated four times for 10 min (30 s on mode, 30 s off mode) using the Bioruptor Plus (Diagenode, Belgium). Sheared chromatin was collected by centrifugation and a 10-μg sample was used for IP. Following pre-clearing for 2 h at 4 °C, chromatin was incubated in 500 μl with 5 μg of specific antibodies against CtBPs (#sc-17805, Santa Cruz, CA, US), EZH2 (#C15410039, Diagenode, Belgium), H3K27me3 (#C15410069, Diagenode, Belgium), mouse IgG (#C15400001, Diagenode, Belgium) or rabbit IgG (#C15410206, Diagenode, Belgium) overnight with gentle rotation at 4 °C. Antibodies were validated in accordance to manufacturer's protocols. Recombinant magnetic protein G beads (Invitrogen, MA, USA) were added for 1 h at 4 °C. Afterwards, the beads were washed with low-salt buffer 1 (20 mM Tris/HCl (pH 8.2), 10 mM EDTA, 150 mM NaCl, 0.1% SDS, 1% Triton X-100), high salt buffer 2 (20 mM Tris/HCl (pH 8.2), 10 mM EDTA, 500 mM NaCl, 0.1% SDS, 1% Triton X-100), buffer 3 (10 mM Tris/HCl (pH 8.1),10 mM EDTA, 1% Igepal, 1% deoxycholate, 250 mM LiCl) and 3 times with Tris-EDTA buffer (10 mM Tris/HCl (pH 8.0), 1 mM EDTA). Each wash step was performed for 10 min at 4 °C. Samples were eluted in 100 μl elution buffer containing 0.1% SDS, 0.1 M NaHCO$_3$, (pH 8.0) for 1 h at RT. To reverse cross-linking, samples were supplemented with 0.2 M NaCl and incubated overnight at 65 °C. Proteinase K was added at a final concentration of 0.5 μg/μl for 2 h at 42 °C. DNA was purified with a commercial gel and PCR clean-up kit (Macherey-Nagel, Germany) and was used directly for quantitative PCR with SimpleChIP human CDKN1A promoter primers (Cell Signaling Technology, MA, US). ChIP-qPCR data were normalized to input DNA and presented as a percentage of input. P values (Two-sided Student's *t*-test) were calculated based on at least three biological replicates.

## In silico validation

*RAI2* mRNA expression was analyzed in the prostate adenocarcinoma cohort of the TCGA PanCancer Atlas at the cBIO portal[23,89]. The co-expression tab was used for correlative analysis with *EZH2* gene expression.

## Gene expression analysis in circulating tumor cells (CTC)

A total of 117 blood samples were obtained from 92 patients with PC, who were treated at the Center of Oncology, University Medical Center Hamburg-Eppendorf from June 2018 to July 2022. Patients were selected by an experienced clinician based on clinical parameters and serum markers according to modified criteria defined by Epstein et al. and Aparicio et al.[29,30]. All blood samples were processed within 3 h of blood collection. From each sample, 7.5 ml of blood was collected into EDTA blood collection tubes and processed within 2 h using the AdnaTest (Qiagen, Hilden, Germany). Patients were divided into HSPC, CRPC, AVPC[29], and NEPC[30], based on their clinical status. Patient information including age, characteristics, CTC presence, and gene expression patterns are listed in Supplementary Data 7. Additionally, blood from 10 healthy donors was obtained from the Institute of Transfusion Medicine, University Medical Center Hamburg-

Eppendorf, as a negative control. Gene expression in CTCs was analyzed by semiquantitative RT-PCR as described[26]. Briefly, whole blood was incubated with immunomagnetic beads directed against epithelial cell markers, and the enriched cells were washed and lysed. mRNA purification and cDNA synthesis were performed using the AdnaTest Prostate Cancer Detect and the Sensiscript RT Kit (Qiagen, Hilden, Germany) according to the manufacturer's instructions. For PCR-based preamplification, 5 μL of cDNA was mixed in a 50 μl reaction with TATAA SYBR® GrandMaster® Mix and GrandPerformance Assays (TATAA Biocentre, Göteborg, Sweden), containing the forward and reverse primers (50 nM, final concentration, respectively) of each specific target gene *AR* (qA-01-0364P), *KRT19* (qA-01-0225P), *RAI2* (qA-01-0890P) and the ValidPrime® Assay (TATAA Biocenter, Göteborg, Sweden), to test for the presence of gDNA in the samples. The CFXTM Real Time System (Bio-Rad, CA, USA) was set to the following pre-amplification program: initial denaturation at 95 °C for 3 min, followed by 18 cycles of denaturation at 95 °C for 15 s, annealing at 60 °C for 2 min and elongation at 72 °C for 1 min. Preamplification reactions were diluted 1:8 and subjected to individual qPCR analysis per assay. Missing values were replaced by ΔCq max +2 on a gene-specific basis. All ΔCq values higher than the mean ΔCq + 1 SD of the healthy donors were also replaced. Samples were considered positive for the expression of the respective gene if a ΔCq value below the expression in healthy donors was calculated. Samples positive for *AR* or *KRT19* or for both were defined as CTC positive. Fisher's exact test was used for statistical testing.

## Statistics and reproducibility

Relevant information on statistical data analysis is provided in the legends of the respective figure panels. Unless otherwise stated, experiments were performed in biological replicates. Statistical analysis was performed using GraphPad Prism 10.0. (Dotmatics, MA, USA).

## Reporting summary

Further information on research design is available in the Nature Portfolio Reporting Summary linked to this article.

# Data availability

The coordinates of the X-ray structure determined in this contribution have been deposited in the Protein Data Bank under accession codes 8ATI. Chemical shifts of RAI2(WT, 303-465) were assigned and deposited in the BMRB under accession number 28085. The coordinates of the cryogenic-EM structure have been deposited in the protein data bank under accession code 8ARI and the density map has been deposited in the Electron Microscopy Data Bank (EMDB) under the accession code EMD-15603. The SAXS data have been deposited at the SASBDB server under accession codes: SASDQW5, SASDQZ5, SASDQ26, SASDQ36, SASDQ46, SASDQ56, SASDQ66, SASDQ76, SASDQ86, SASDQ96, SASDQA6, SASDQB6, SASDQC6, SASDQD6, SASDQE6, and SASDQF6. The following datasets were reused: the structure coordinates from the PDB for human CtBP1 6CDF and the structure coordinates from the PDB for human CtBP2 dehydrogenase complexed with NAD(H) 2OME. Source data are provided with this paper.

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

## Acknowledgements

The SAXS and X-ray diffraction data were collected at beamlines P12 and P13 operated by the EMBL Hamburg Unit at the PETRA III storage

ring (DESY, Hamburg, Germany), respectively. We also acknowledge access to the EM facility at the Centre for Structure Systems Biology, Hamburg, for sample screening and collection. We acknowledge technical support from the SPC facility at EMBL Hamburg, the Proteomics Core Facility at EMBL Heidelberg, the University Medical Center Hamburg-Eppendorf Microscopy Imaging Facility (DFG Research Infrastructure Portal: RI_00489), and the FACS Sorting Core Unit. We thank Eric Metzger from the University Freiburg Medical Center for help in setting up the chromatin-binding experiments. We thank Bettina Steinbach for technical assistance and Katharina Besler from the University Medical Center Hamburg-Eppendorf for project management support. We also thank our EMBL colleagues Annabel Parret, Kyung Min Noh, Christian Tischer, Cy Jeffries, and Toby Gibson, and Caitlin MacCarthy from the Max Planck Institute for Molecular Biomedicine, Münster, Germany, for technical and scientific advice at various stages of the project. This project has been supported by Deutsche Forschungsgemeinschaft (DFG 218826742 to H.W., K.P., and M.W.), (SPP 2084: µBONE to K.P. and H.W.), (WE 5844/5-1 to S.W. and 341/25-1 to K.P.) and by Deutsche Krebshilfe (70113304) to S.W.

## Author contributions

N.G., S.W., A.L., O.O., G.vA., K.P., and M.W. designed the project. N.G., S.W., E.M., S.G., L.B., A.L., A.W., S.S., and G.vA. performed the experiments. N.G., S.W., E.M., S.G., L.B., H.M., S.S., G.C., O.O., G.vA., and M.W. analyzed the data. N.G., S.W., E.M., O.O., and M.W. wrote the manuscript. S.W., H.W., G.vA., K.P., and M.W. supported the project.

## Funding

## Competing interests

The authors declare no competing interests.
