## [Peer Review File · Nature Communications]

Master corepressor inactivation through multivalent SLiM-induced polymerization mediated by the oncogene suppressor RAI2REVIEWER COMMENTS

Reviewer #1 (Remarks to the Author):

Goradia et al show that the transcriptional master repressor CtBP, which forms tetramers, can polymerize upon binding to multiple sLiMs in the metastasis suppressor RAI2. Co-polymerization of the two leads to the sequestration of CtBP in nuclear foci, which results in de-repression. One of the strengths of this manuscript is the integration of a large set of techniques for structural, biochemical and function characterization. It is overall well written, and the message is clear. One thing that should be corrected is the strawman argument that little is known about how IDRs interact with and regulate their folded binding partners. There is plenty of literature in this area, and I will point to two papers here as examples (PMID: 22398450, PMID: 30244836). The overly simplifying statements to this effect in the Abstract and Discussion should be removed. The Discussion mentions that particularly sLiMs are not well understood, and I disagree with this; sLiMs are often very clearly visible in structures of complexes and studied extensively in the context of short peptides. Overall, the manuscript makes interesting conceptual advancements and is also of high value for the particular system under study.

It is not full clear from the Discussion how the polymerization reliefs transcriptional repression. Is the idea that this is achieved via sequestration of the repressor away from its normal binding sites?

There is some disconnect between the NMR data presented in the figure and the accompanying text. While the text mentions random coil chemical shift, they are not shown in the figures. Instead, NMR parameters are reported that are never mentioned in the text, including hetNOE values. This should be corrected.

The authors say that the interfaces between filament layers do not contain specific interactions. What does this mean exactly? Does this mean that the individual layers will stick to many other proteins? In any case, Figs. 3B,C, which are pointed to for this statement fail to make the point. What do the authors really mean and can this be shown better?

Fig. 1H: Not clear that binding to ALDLS-1 and -2 is specific. From the NMR data it looks as if there are additional motifs N-terminal of ALDLS-1 and between motifs 1 and 2. This may lead to reinterpretation of other data.

Fig. 2A and ED Fig. 6C seem to plot the same data (i.e., SEC profile of CtBP1 and CtBP1/RAI complexes), but they look different. In Fig. 2A, binding of RAI2 mutants results in a small decrease of the elution volume, in ED Fig. 6C, the elution volumes are identical.

Minor comments:

ED Fig. 3: The y-axis labeling is poor. I assume Lg means log. The large drop at small q points to aggregation.

Axis labeling is missing or insufficient in Fig. 1 G, H

Given that the text mentions KD values, it would be nice to also report KD instead of KA values in Fig. 1D.

"... supporting a protein environment independent nature of the RAI2/CtBP1 interaction as separate sLiMs." It is unclear to me what this means.

"Given that both protein binding partners have multivalent binding sites": The authors likely mean that both proteins have multiple binding sites or that they are multivalent for each other.

"In contrast, addition of the M1 and M2 variants to CtBP1 resulted in an increase in retention volume..." In fact, addition of M1 and M2 variants to CtBP1 resulted in a decrease in retention volume. Just relative to the complex with WT RAI2, the retention volume is increased.

Fig. 2C is shown too small to see any details. I cannot see foci.

"we first performed an in vitro transactivation assay in human embryonic kidney (HEK) 293T cells..." In vitro means in cells here?

Does genotoxic stress raise RAI2 levels?

Reviewer #2 (Remarks to the Author):

The manuscript by Goradia et al., described the detailed characterization of the multivalent SLiM-based interactions between RAI2 and CtBPs, which induces CtBP polymerization. The authors are linking the results to different diseases including cancer as RAI2 is a tumor suppressor and CtBPs are oncogenes. Also, viral CtBP binding motifs are competing for the same site. I very much enjoyed reading the manuscript, as it is well written and as far as I can see based on solid experimental results.

I only have one minor comment. In the abstract, the author call the motifs "RAI2-like SLiM motifs". To avoid confusion, I advice the authors to call it a variant of CtBP binding PxDSL motif, which is already defined and listed in the ELM database, and which the authors refer to in the text.

Reviewer #3 (Remarks to the Author):

Goradia et al. NCOMMS-23-56116-T

Master corepressor inactivation through multivalent SLiM-induced polymerization mediated by the oncogene suppressor RAI2

The transcriptional master corepressor CtBP binds the putative metastasis suppressor RAI2 through recognition of short linear sequence motifs, such as the PxDSL motif. Goradia et al. exploit a notable array of integrated biophysical, structural and cell biology methods to characterize such interaction (the role of transcriptional repressors in cancer treatment is an active area of research, hence the added relevance of this study). The amount of work and results presented are impressive, stemming from a collaboration between well-established and large institutions. All the biophysical methods applied are state of the art and properly exploited.

The study shows that, in the presence of the metastasis suppressor protein RAI2, tetramers of CtBP assemble into elongated fibrils endowed with a specific overall superstructure that is based on the association of CtBP2 tetramers. As a result, RAI2 induced CtBP nuclear foci and, importantly, induced CtBP corepressor loss of function, providing a clear example of regulation through an intrinsically disordered protein (RAI2) that triggers CtBP polymerization (i.e. target deactivation through polymerization). The manuscript subsequently presents cell biology studies that characterize loss of EZH2 catalytic activity in the presence of CtBP polymerization, and an analysis of RAI2 gene expression in the context of prostate cancer progression. Implications for therapeutic intervention in cancer treatment are discussed, together with general conclusions on the functional roles played by intrinsically disordered proteins in their interaction-mediated regulatory events.

I find this communication solid, extensive for the breadth of experimental approaches exploited, and significant for the different fields of interest covered (unfolded protein recognition and interactions, oncogene regulation mechanism, tumor suppressor regulation, molecular bases of cancer). I must also admit that, given my specific competence, I cannot be critical enough on the cell biology experiments here reported, which nevertheless complement properly the mechanistic principles discovered in the first part; thus, this review focusses mostly on the biophysical and structural parts of the manuscript.

After examining the manuscript in detail, I recommend this manuscript for publication in Nature

Communications.

Nevertheless, I also recommend solving some conundrums (below) that I spotted while reading.

1) I find that the notations adopted to address the different protein constructs are a bit misleading (throughout text and figures). To be clear, here is an example: RAI2(303-530) clearly stands for a truncated form of RAI2 encompassing residues 303-530. Then, what RAI2(WT, 303-530) means? I may guess that it stands for a truncated form (303-530) of the WT species. The same notation, not always used, applies to other truncated RAI2 forms appearing in the text and in the figures. A simple explanatory text, and a consistent use of a unique notation, would help the reader (recommended).

2) It is important to know the distance (residue wise along sequence) of the two ALDLS peptides in RAI2; it can only be extracted from Fig. 1E; it should be specified in the text in an early part of the manuscript.

3) Extended Data Table 2 presents values of the equilibrium constants for the reaction of CtBP1 with several RAI2 variants. The information provided is obviously relevant. Nevertheless, it appears to me odd to present two equilibrium constants (for direct and reverse reactions), which refer to the same equilibrium, and that numerically are just one the reciprocal of the other. One of the two K values would have been sufficient. Moreover, from the specific section in Materials and Methods (Quantitative determination of binding affinities) the ITC experiments described appear to relate to the measurement of the association equilibrium constant. Explain the reasons for this choice.

4) Extended data Table 2 – The equilibrium constant values for CtBP1(K46W) binding to RAI2(M1/M2, 303-362) appear to be comparable to the values measured for WT CtBP1 binding to the M1 and M2 variants, that associate CtBP1 through just one ALDLS peptide. It is worth mentioning this coincidence in the text, since it would add to the consistency of the mechanisms presented. Nevertheless, the interpreted binding stoichiometries reported in the Table may be different. Any explanation?

5) Page 6 – why the polymerization CtBP/RAI2(WT, 303-465) should yield finite size particles (28 nm, determined by SAXS), instead of extending indefinitely? Indeed, the negative staining images of Extended Data Fig.12 show much longer filaments.

6) Page 6, closing lines at page center: "Our findings for CtBP2/RAI2 ... ", although the sentence can be easily understood in the context, given the contents of Fig.2C-D and Extended Data Fig.7, there seems to be a typo, and the sentence should read: "Our findings for CtBP1/RAI2 assembly ... for CtBP2 ..."

7) Page 6, line 3 from bottom: ... which directly interact with ...

8) Page 7: "... In this structure, the CtBP1/RAI2 fiber is assembled by 322 symmetry ..." I find difficult to follow this symmetry definition, since 322 would represent 'point symmetry' that hardly applies to a fiber. From what I read, the fiber symmetry could be described by a 32 screw axis (or 31). A more thorough explanation is needed.

9) For similar reasons (Fig.3 caption), I find a bit misleading the related sentence "... a 2-fold repeated ALDS motif (orange) ...": firstly, correct the peptide sequence (ALDLS); then, the sentence seems to imply a 2-fold symmetry relationship between ALDLS(1) and ALDLS(2); I would suggest "... the twice repeated ALDLS motif (orange) ...".

10) Also (Fig.3 caption), a "... Book-like opening of a CtBP1(n)/CtBP1(n+1) interface", to be precise, cannot encompass the structure of the RAI2 polypeptide bound at the interface, therefore a few more explanatory words are needed.

11) Page 8, line 4: ... that directly connects ...

Martino Bolognesi

POINT-TO-POINT RESPONSE ON REVIEWER COMMENTS

The comments addressed have been numbered for the sake of clarity and cross-referencing. All replies inserted are in grey-blue both in the manuscript text and point-to-point response.

Reviewer #1 (Remarks to the Author):

Goradia et al show that the transcriptional master repressor CtBP, which forms tetramers, can polymerize upon binding to multiple sLiMs in the metastasis suppressor RAI2. Co-polymerization of the two leads to the sequestration of CtBP in nuclear foci, which results in de-repression. One of the strengths of this manuscript is the integration of a large set of techniques for structural, biochemical and function characterization. It is overall well written, and the message is clear.

1.1. One thing that should be corrected is the strawman argument that little is known about how IDRs interact with and regulate their folded binding partners. There is plenty of literature in this area, and I will point to two papers here as examples (PMID: 22398450, PMID: 30244836). The overly simplifying statements to this effect in the Abstract and Discussion should be removed. The Discussion mentions that particularly sLiMs are not well understood, and I disagree with this; sLiMs are often very clearly visible in structures of complexes and studied extensively in the context of short peptides.

It was in no way our intention to downplay the seminal contributions in the field of intrinsically disordered proteins by any means, and in fact we had extensively exchanged on this matter with internal experts in the field (Toby Gibson, Balint Meszaros) and we are also aware about the extensive literature. To avoid any unintended impression of inferiority we have removed the expression "... lagging behind..." and have modified the first sentence of the abstract as follows: *"While the elucidation of regulatory mechanisms of folded proteins is facilitated due to their amenability to high-resolution structural characterization, investigation of these mechanisms in disordered proteins is more challenging due to their structural heterogeneity, which can be captured by a variety of biophysical approaches."*

Similarly, we have modified the following sentence in the Discussion section: *"Those involving mostly folded proteins such as allosteric processes, changes in assembly state and post-translational modifications are well established, but insight into molecular mechanisms involving unfolded proteins is more challenging and requires innovative approaches with biophysical methods, to provide structural insight at low resolution."*

If the reviewer wishes, we are open to further changes, as long as they do not affect the presentation of the experimental data in this contribution. Once again, it has not been our intention to devalue amazing data from the intrinsically disordered proteins by any means. As this is an original research paper, we also wanted to avoid any impression of reviewing or qualifying the field.

Overall, the manuscript makes interesting conceptual advancements and is also of high value for the particular system under study.

1.2. It is not full clear from the Discussion how the polymerization reliefs transcriptional repression. Is the idea that this is achieved via sequestration of the repressor away from its normal binding sites?

The answer to this question is yes. More details can be found in references 13 and 40, which are cited when this topic is discussed. For further clarification, we have added the following sentence in the Discussion section: *"In this model, CtBP polymerization would counteract tetrameric CtBP as the functionally active assembly."* Because we do not yet know the precise mechanisms of impairing interactions with relevant protein partners at the level of the tetrameric CtBP assembly, we have not engaged into a more in-depth discussion at this point.

1.3. There is some disconnect between the NMR data presented in the figure and the accompanying text. While the text mentions random coil chemical shift, they are not shown in the figures. Instead, NMR parameters are reported that are never mentioned in the text, including hetNOE values. This should be corrected.

The random coil chemical shifts of the full wt [1H,15N]-HSQC fingerprint spectra of three RAI2(303-362) variants (WT, M1, M2) are shown in the Extended Data Fig. 5 and are listed in the Extended Data Table 1, which is referenced by the following statement in the main text: "*In addition, nuclear magnetic resonance (NMR) spectroscopy chemical shift analysis of RAI2(303-362) revealed typical random coil values throughout the entire sequence of this protein fragment (Fig. 1E-F, Extended Data Fig. 5, Extended Data Table 1).*"

We have added the following statement: "*Evaluation of the molecular flexibility (Extended Data Fig. 5d) revealed that all RAI2 residues exhibited a HN bond mobility on the fast time scale, which are reflected by negative 15N-[1H]-NOE values.*"

1.4. The authors say that the interfaces between filament layers do not contain specific interactions. What does this mean exactly? Does this mean that the individual layers will stick to many other proteins? In any case, Figs. 3B,C, which are pointed to for this statement fail to make the point. What do the authors really mean and can this be shown better?

We have made the following text modification: "*..., except for isolated hydrogen bonds involving residues K46 and R336 with the next CtBP layer (Fig. 3B-C). This may explain why CtBP filament formation requires the presence of RAI2 as a polymerization mediator.*" The intent of this change is to improve pointing to the content especially of Fig. 3c.

1.5. Fig. 1H: Not clear that binding to ALDLS-1 and -2 is specific. From the NMR data it looks as if there are additional motifs N-terminal of ALDLS-1 and between motifs 1 and 2. This may lead to reinterpretation of other data.

It is a well-known observation in NMR mapping experiments that the chemical shifts of neighboring stretches are often also affected by the binding process (Becker *et al.* (2018) *Chemphyschem* 19(8):895-906. doi: 10.1002/cphc.201701253). In this contribution, especially the RAI2 mutant data show the relevance of the two ALDLS motifs in the CtBP interaction. This finding is further corroborated by the stoichiometry data presented in Fig. 1d.

1.6. Fig. 2A and ED Fig. 6C seem to plot the same data (i.e., SEC profile of CtBP1 and CtBP1/RAI2 complexes), but they look different. In Fig. 2A, binding of RAI2 mutants results in a small decrease of the elution volume, in ED Fig. 6C, the elution volumes are identical.

ED Figure 6c shows a negative control experiment, using RAI(M1+M2), in which both CtBP binding motifs M1 and M2 were impaired. As expected, since there is no binding of RAI2(M1+M2) to CtBP, there was no shift in the SEC elution profile. For reasons of clarity, this experiment is not shown in Figure 2a. Separate CtBP is indeed shown in both figures, as a reference for assessing any shifts of the elution volume due to RAI2 binding.

Minor comments:

1.7. ED Fig. 3: The y-axis labeling is poor. I assume Lg means log. The large drop at small q points to aggregation.

The y-axis label has been changed from Lg to Log. What actually is observed is an increase in scattering intensity at very low scattering angles, due to aggregated fractions caused by rapid radiation damage of the sample (for further details see: Jeffries *et al.* (2015) *J Synchrotron Radiat.* 22(2):273-

9. doi: 10.1107/S1600577515000375.). We also have discussed this with the first author of the paper (Cy Jeffries).

1.8. Axis labeling is missing or insufficient in Fig. 1 G, H

Complete axis labels have been added.

1.9. Given that the text mentions KD values, it would be nice to also report KD instead of KA values in Fig. 1D.

The revised Fig. 1d depicts KD instead of KA values. The figure caption has been modified accordingly.

1.10 "... supporting a protein environment independent nature of the RAI2/CtBP1 interaction as separate SLiMs." It is unclear to me what this means.

We have modified the following sentence: "..., demonstrating that the nature of the RAI2/CtBP1 interaction through twofold repeated SLiMs from RAI2 does depend on other parts beyond the sequence of the smallest RAI2 construct (303-362) used for these experiments (**Extended Data Table 2**)."

1.11. "Given that both protein binding partners have multivalent binding sites": The authors likely mean that both proteins have multiple binding sites or that they are multivalent for each other.

We have modified the relevant sentence for the sake of improved clarity: "Given that both protein binding partners have multivalent binding sites for each other, four in CtBP and two in RAI2, we wondered about the possibility of assembly-mediated oligomerization."

1.12. "In contrast, addition of the M1 and M2 variants to CtBP1 resulted in an increase in retention volume..." In fact, addition of M1 and M2 variants to CtBP1 resulted in a decrease in retention volume. Just relative to the complex with WT RAI2, the retention volume is increased.

To rule out any ambiguity in this statement, we have modified the relevant sentence: "In contrast, addition of the M1 and M2 variants to CtBP1 resulted in an increase in retention volume relative to the RAI2(WT, 303-465)/CtBP elution profile, suggesting non-oligomerized, lower molecular weight CtBP1/RAI2 complexes (**Fig. 2a, Extended Data Table 4**)."

1.13. Fig. 2C is shown too small to see any details. I cannot see foci.

Fig. 2c has been enlarged to enhance the visibility of the foci.

1.14. "we first performed an in vitro transactivation assay in human embryonic kidney (HEK) 293T cells..." In vitro means in cells here?

Transactivation assays are generally considered as "in vitro" assays as they do not use a readout from an endogenous interaction and hence are limited in terms of functional interpretation. We confirm that these experiments have been carried out in cells.

1.15. Does genotoxic stress raise RAI2 levels?

Genotoxic stress does indeed increase RAI2 gene expression and protein levels in breast cancer cells. For further details see: <https://doi.org/10.1158/1538-7445.AM2018-3364>.

Reviewer #2 (Remarks to the Author):

The manuscript by Goradia et al., described the detailed characterization of the multivalent SLiM-based interactions between RAI2 and CtBPs, which induces CtBP polymerization. The authors are linking the results to different diseases including cancer as RAI2 is a tumor suppressor and CtBPs are oncogenes. Also, viral CtBP binding motifs are competing for the same site. I very much enjoyed reading the manuscript, as it is well written and as far as I can see based on solid experimental results.

2.1. I only have one minor comment. In the abstract, the author call the motifs “RAI2-like SLiM motifs”. To avoid confusion, I advice the authors to call it a variant of CtBP binding PxDSL motif, which is already defined and listed in the ELM database, and which the authors refer to in the text.

In the abstract, we introduce the concept repetitive Short Linear Sequence Motifs (SLiMs) as a driver of the observed polymerization of RAI2-mediated polymerization of CtBP as a key underpinning mechanistic principle. We have avoided using the term PxDSL in the Abstract, as it has not been introduced at this point and the length limitations of the Abstract do not provide sufficient space for it. This has subsequently been done in the introduction. If the referee has a very strong opinion along these lines, we would be open to any alternatives.

Reviewer #3 (Remarks to the Author):

Goradia et al. NCOMMS-23-56116-T

Master corepressor inactivation through multivalent SLiM-induced polymerization mediated by the oncogene suppressor RAI2

The transcriptional master corepressor CtBP binds the putative metastasis suppressor RAI2 through recognition of short linear sequence motifs, such as the PxDSL motif. Goradia et al. exploit a notable array of integrated biophysical, structural and cell biology methods to characterize such interaction (the role of transcriptional repressors in cancer treatment is an active area of research, hence the added relevance of this study). The amount of work and results presented are impressive, stemming from a collaboration between well-established and large institutions. All the biophysical methods applied are state of the art and properly exploited.

The study shows that, in the presence of the metastasis suppressor protein RAI2, tetramers of CtBP assemble into elongated fibrils endowed with a specific overall superstructure that is based on the association of CtBP2 tetramers. As a result, RAI2 induced CtBP nuclear foci and, importantly, induced CtBP corepressor loss of function, providing a clear example of regulation through an intrinsically disordered protein (RAI2) that triggers CtBP polymerization (i.e. target deactivation through polymerization). The manuscript subsequently presents cell biology studies that characterize loss of EZH2 catalytic activity in the presence of CtBP polymerization, and an analysis of RAI2 gene expression in the context of prostate cancer progression. Implications for therapeutic intervention in cancer treatment are discussed, together with general conclusions on the functional roles played by intrinsically disordered proteins in their interaction-mediated regulatory events.

I find this communication solid, extensive for the breadth of experimental approaches exploited, and significant for the different fields of interest covered (unfolded protein recognition and interactions, oncogene regulation mechanism, tumor suppressor regulation, molecular bases of cancer). I must also admit that, given my specific competence, I cannot be critical enough on the cell biology experiments here reported, which nevertheless complement properly the mechanistic principles discovered in the first part; thus, this review focusses mostly on the biophysical and structural parts of the manuscript.

After examining the manuscript in detail, I recommend this manuscript for publication in Nature Communications. Nevertheless, I also recommend solving some conundrums (below) that I spotted while reading.

3.01. I find that the notations adopted to address the different protein constructs are a bit misleading

(throughout text and figures). To be clear, here is an example: RAI2(303-530) clearly stands for a truncated form of RAI2 encompassing residues 303-530. Then, what RAI2(WT, 303-530) means? I may guess that it stands for a truncated form (303-530) of the WT species. The same notation, not always used, applies to other truncated RAI2 forms appearing in the text and in the figures. A simple explanatory text, and a consistent use of a unique notation, would help the reader (recommended).

We have implemented a consistent nomenclature of the type “RAI2(WT, 303-530)” throughout the complete manuscript.

3.02. It is important to know the distance (residue wise along sequence) of the two ALDLS peptides in RAI2; it can only be extracted from Fig. 1E; it should be specified in the text in an early part of the manuscript.

We have expanded the relevant sentence in the first paragraph of the Results section in the following way: “*As RAI2 remarkably contains two CtBP-binding motifs of identical sequence (ALDLS) at RAI sequence positions 316-320 and 342-346, respectively, thus separated by a short 21-residue linker only, we first investigated whether there is preferential binding by one of the two ALDLS motifs through a specific protein environment and whether there is any interference between the two of them.*”

3.03. Extended Data Table 2 presents values of the equilibrium constants for the reaction of CtBP1 with several RAI2 variants. The information provided is obviously relevant. Nevertheless, it appears to me odd to present two equilibrium constants (for direct and reverse reactions), which refer to the same equilibrium, and that numerically are just one the reciprocal of the other. One of the two K values would have been sufficient. Moreover, from the specific section in Materials and Methods (Quantitative determination of binding affinities) the ITC experiments described appear to relate to the measurement of the association equilibrium constant. Explain the reasons for this choice.

This has been corrected. See also reply to query 1.9.

3.04. Extended data Table 2 – The equilibrium constant values for CtBP1(K46W) binding to RAI2(M1/M2, 303-362) appear to be comparable to the values measured for WT CtBP1 binding to the M1 and M2 variants, that associate CtBP1 through just one ALDLS peptide. It is worth mentioning this coincidence in the text, since it would add to the consistency of the mechanisms presented. Nevertheless, the interpreted binding stoichiometries reported in the Table may be different. Any explanation?

The relevant sentence has been modified: “*The binding affinity of CtBP1(K46W) with RAI2 was significantly decreased, similar to values observed for CtBP(WT) and RAI2(M1) or RAI2(M2) mutants (Extended Data Table 2), which could indicate reduced polymerization and fragmented CtBP/RAI2 fiber formation.*” In the revised version, the argument about fragmented fiber formation is directly linked to the observed changes in K_D , as opposed to changes in stoichiometry in the original version. To respond to the question about a possible explanation, our guess would be that this could be due to diminished accessibility of RAI2 binding sites within more disordered CtBP/RAI2 filaments, as indicated in the Extended Data Fig. 12. However, as we have no validated experimental evidence for this, we prefer to avoid expressing any speculative arguments in the manuscript.

3.05. Page 6 – why the polymerization CtBP/RAI2(WT, 303-465) should yield finite size particles (28 nm, determined by SAXS), instead of extending indefinitely? Indeed, the negative staining images of Extended Data Fig. 12 show much longer filaments.

We are aware of this discrepancy. In our view, this is due to quite different experimental conditions in SAXS and negative staining EM experiments. It should be noted that protein filtering from aggregates was required to obtain interpretable SAXS data, implying that the sample used for SAXS experiments was presumably enriched in shorter, non-aggregated CtBP/RAI2 filaments. See also reply to point 1.7.

3.06. Page 6, closing lines at page center: “Our findings for CtBP2/RAI2 ... “, although the sentence can be easily understood in the context, given the contents of Fig.2C-D and Extended Data Fig.7, there seems to be a typo, and the sentence should read: “Our findings for CtBP1/RAI2 assembly for CtBP2 ...”

We have modified the sentence: “Our findings for CtBP2/RAI2 assembly are consistent with those observed for CtBP1/RAI2 (**Extended Data Fig. 7B-C**).”

3.07. Page 6, line 3 from bottom: ... which directly interact with ...

This has been corrected. The modified sentence reads: “*Apart from RAI2 residues 315-321, which directly interact with CtBP2 (Fig. 2E, right panel, Extended Fig. 10B, right panel), ...*”

3.08. Page 7: “... In this structure, the CtBP1/RAI2 fiber is assembled by 322 symmetry ...” I find difficult to follow this symmetry definition, since 322 would represent ‘point symmetry’ that hardly applies to a fiber. From what I read, the fiber symmetry could be described by a 32 screw axis (or 31). A more thorough explanation is needed.

The relevant sentences have been modified in the following way: “*In this structure, the CtBP1/RAI2 fiber is assembled by a 3-fold axis that defines the longitudinal fiber axis and coincides with one of the 2-fold axes of the CtBP1 tetrameric assembly (Fig. 2E, Fig. 3A). This arrangement is rotated by 120 degrees and translated by 5 nm for adjacent CtBP1 tetrameric layers, adding a 3₁ screw component to the 3-fold filament axis.*”

3.09. For similar reasons (Fig.3 caption), I find a bit misleading the related sentence “... a 2-fold repeated ALDS motif (orange) ...”: firstly, correct the peptide sequence (ALDLS); then, the sentence seems to imply a 2-fold symmetry relationship between ALDLS(1) and ALDLS(2); I would suggest “... the twice repeated ALDLS motif (orange) ...”.

To avoid confusion, we have modified the relevant sentence in the following way: “*Adjacent CtBP1 layers are rotated around a vertical 3-fold filament axis and are connected by a pair of RAI2 peptides (yellow) with an ALDS tandem motif (orange) in opposite positions to the central filament axis.*”

3.10. Also (Fig.3 caption), a “... Book-like opening of a CtBP1(n)/CtBP1(n+1) interface”, to be precise, cannot encompass the structure of the RAI2 polypeptide bound at the interface, therefore a few more explanatory words are needed.

The relevant sentence has been modified: “*Book-like opening of a CtBP1(n)/CtBP1(n+1) interface (boxed) without bound RAI2 is indicated schematically.*”

3.11. Page 8, line 4: ... that directly connects ...

The relevant sentence has been corrected: “*Within this arrangement, we observed two RAI2 peptides per CtBP1/CtBP1 layer interface that directly connects the RAI2-binding sites of adjacent CtBP1 tetrameric layers via their two ALDLS motifs 1 and 2*”.

REVIEWERS' COMMENTS

Reviewer #1 (Remarks to the Author):

I will reiterate my appreciation of the breadth of techniques used to comprehensively study this complex system and extract structure, mechanism and function. I further appreciate the authors' explanations and edits regarding my comments. These are satisfactory. the explanation that the high scattering intensity at low q in some of their samples is likely caused by radiation damage should be acknowledged in the text or figure caption.
This is a beautiful piece of work and I support publication.

Reviewer #2 (Remarks to the Author):

The authors have satisfactorily replied to my minor comment.

Reviewer #3 (Remarks to the Author):

The manuscript by Goradia et al. (resubmitted) was amended thoroughly, answering 11 issues I raised in my previous review, but also providing answers that intertwine with comments raised in parallel by two different reviewers. In the present version the manuscript is complete and clearly presented. The topic dealt with is front-line in the field of gene expression regulation in cancer. The figures present correctly the concepts and the many biochemical/biophysical methods and are thoroughly readable. There is plenty of supplementary information to complement the main text. As a whole I recommend the current version for publication in Nature Communications .
Martino Bolognesi